# Strong Hydrogen Bonds in Acetylenedicarboxylic Acid Dihydrate

**DOI:** 10.3390/ijms23116164

**Published:** 2022-05-31

**Authors:** Urban Novak, Amalija Golobič, Natalija Klančnik, Vlasta Mohaček-Grošev, Jernej Stare, Jože Grdadolnik

**Affiliations:** 1National Institute of Chemistry Ljubljana, Hajdrihova 19, 1000 Ljubljana, Slovenia; urban.novak@ki.si (U.N.); jernej.stare@ki.si (J.S.); 2Faculty of Chemistry and Chemical Technology, University of Ljubljana, Večna pot 113, 1000 Ljubljana, Slovenia; amalija.golobic@fkkt.uni-lj.si (A.G.); natalija.klancnik@fkkt.uni-lj.si (N.K.); 3Ruđer Bošković Institute, Bijenička c. 54, 10002 Zagreb, Croatia; vlasta.mohacek.grosev@irb.hr

**Keywords:** strong hydrogen bond, acetylenedicarboxylic acid dihydrate, vibrational spectroscopy, X-ray structure, QM calculations, OH stretching band

## Abstract

Acetylenedicarboxylic acid dihydrate (ADAD) represents a complex with strong hydrogen bonding between the carboxylic OH and the water molecule. An X-ray re-examination of the ADAD crystal structure confirms the O^…^O distance of the short hydrogen bonds, and clearly shows different bond lengths between the two oxygen atoms with respect to the carbon atom in the carboxyl group, indicating a neutral structure for the complex. The neutral structure was also confirmed by vibrational spectroscopy, as no proton transfer was observed. The diffraction studies also revealed two polymorph modifications: room temperature (α) and low temperature (β), with a phase transition at approximately 4.9 °C. The calculated vibrational spectra are in satisfactory agreement with the experimental spectra. A comparison of the structure and the vibrational spectra between the ADAD and the oxalic acid dihydrate reveals some interesting details. The crystal structures of both crystal hydrates are almost identical; only the O^…^O distances of the strongest hydrogen bonds differ by 0.08 Å. Although it was expected that a larger O^…^O spacing in the ADAD crystal may significantly change the infrared and Raman spectra, especially for the frequency and the shape of the acidic OH stretching vibration, both the shape and frequency are almost identical, with all subpeaks topped on the broad OH stretching vibration. The O^…^O distance dependent are only in- and out-of-plane OH deformations modes. The presence of polarons due to the ionized defects was not observed in the vibrational spectra of ADAD. Therefore, the origin of the broad OH band shape was explained in a similar way to the acid dimers. The anharmonicity of a potential enhances the coupling of the OH stretching with the low-frequency hydrogen bond stretching, which, in addition to the Fermi resonance, structures the band shape of the OH stretching. The fine structure found as a superposition of a broad OH stretching is attributed to Davydov coupling.

## 1. Introduction

Hydrogen bonds in biological systems are crucial for structure, dynamics, and biological function. Among all of the types of hydrogen bonds, the short (strong) hydrogen bonds are of particular interest. The study of the short hydrogen bonds was strongly motivated when Cleland and Kreevoy proposed that this type of hydrogen bonding could play a central role in certain enzymatic reactions [1]. Their proposal has been extensively studied, both theoretically and experimentally. However, studies of this nature and the role of strong hydrogen bonds have always been somewhat limited, because of the limited number of examples that are available, and the often difficult task of interpreting observations, such as the assignment of infrared spectral bands or the precise localization of the proton in a hydrogen bond. Furthermore, an exact single crystal structure of a hydrogen-bonded system can significantly alleviate these problems, and would serve as an excellent starting point for spectroscopic and theoretical studies. A representative system with short hydrogen bonds is acetylenedicarboxylic acid dihydrate (ADAD).

The crystal and molecular structure of ADAD was determined 75 years ago through X-ray diffraction using photographic film detection [2] in the series of dicarboxylic acid hydrates such as diacetylenedicarboxylic acid [3] and oxalic acid dihydrates [4,5]. In contrast, crystals of the former compound were the subject of numerous X-ray and inelastic neutron scattering diffraction studies [6], and Raman [7,8] and infrared investigations [9,10,11,12,13,14], as well as molecular dynamics modelling [15] can give no indication of a corresponding interest in ADAD. The above crystals have a fairly similar structure, consisting of spiral chains with alternating strong hydrogen bonds between the hydroxyl groups of the acid and the water molecules, and weak hydrogen bonds between the water molecules and the carbonyl group of the acid molecules. In all these crystals, the water molecules form additional weak hydrogen bonds with other acid molecules with the same connectivity of the hydrogen bonding network. However, the structure of ACAD differs with respect to oxalic acid dihydrate in the O^…^O distances, and thus it offers a way of demonstrating the effects of hydrogen bond strength on the vibrational spectra of both. Note that this example is almost free of side effects due to the differences in the environment. Furthermore, the spectra of both acids allow for a clear assignment of the bands formed by the bending of the hydroxyl groups of the strongly hydrogen-bonded acid. It is also interesting to note that a similar structure of hydrogen bonds between water molecules and carboxyl groups was found in the crystal structure of pentadecafluoro octanoic acid hydrate (PDFO) [16]. The O^…^O length of the shortest hydrogen bond is the same as in the crystal of oxalic acid dihydrate (OADH, 2.49 Å, at room temperature). Although the infrared spectra are very similar, especially the pattern that originates in the hydrogen bond network [7,14,17], the structure of the acid molecule is quite different, since PDFO formed a bilayer structure. The hydrogen bond pattern is centrosymmetric, with two water molecules and two carbonyl oxygen atoms forming a planar rhombic arrangement, with the water-hydrogen atoms almost within the plane. The water molecules are connected with weak hydrogen bonds, and they serve as acceptors for the short hydrogen bond with the acidic hydroxyl group.

Generally, in molecular systems forming strong hydrogen bonds with asymmetric potential with an O^…^O spacing between 2.5 Å and 2.6 Å, the O-H stretching band has a typical position between 1500 cm^−1^ and 2000 cm^−1^, and a bandwidth exceeding 500 cm^−1^, and it often shows a fine structure due to Fermi resonances with overtone and combination levels, and Franck–Condon progressions due to anharmonic coupling to the low-frequency hydrogen bonding modes [8,18,19]. The appearance of crystal defects may lead to the formation of polarons, which can also alter the shape of the broad OH stretching absorption [20]. Furthermore, in systems with multiple hydrogen bonds, excitonic or Davydov coupling between different resonant oscillators that are related to hydrogen bonding can occur. However, the most characteristic pattern that is observed in the infrared spectra of strong hydrogen bonds is the presence of a triplet structure of the broad OH band. The sub-bands are generally referred to as ABC bands in the infrared spectra of sulfinic acid [21] and organofosforinic acid [22] after a first detailed characterization. Ab-initio calculations showed that the origin of the structured OH stretching band lies in the anharmonic coupling between the OH and O^…^O stretching modes [23,24], the Davydov coupling between the hydrogen bonds and Fermi resonances of the in-plane, and the out-of-plane OH bending and the combination of these bands [18,19]. Since this is the most commonly used notation of OH stretch structure, we also use it, slightly modified, in our paper.

The present X-ray diffraction results at room temperature are similar to the published ones [2]. However, the later data acquisition based on photographic films was of poor quality, with a large R factor and no location of hydrogen atoms. In our updated results, which also include the positions of the H atoms, the differences in the carbon–oxygen bond lengths of the carboxyl groups are also more pronounced. Moreover, it was reported [25] that ADAD undergoes a reversible phase transition from room temperature (α form) to β form at approximately +3 °C, and an irreversible transition from β to γ form at approximately −40 °C. These conclusions were based only on the measurement of a few reflections using photographic films, and not on calorimetry and structure determination. In this work, a phase transition(s) of ADAD was studied via DSC measurements and the crystal structures of the polymorphic modification at 4.9 °C and near −40 °C were determined via X-ray single crystal structure analysis.

It is considered that deuteration generally does not change crystal structures [26]. On the other hand, there are exceptions where a phenomenon called “isotopic polymorphism” occurs after deuteration [27,28,29,30]. Among the first known examples of this is oxalic acid dihydrate [4]. Recrystallization from D_2_O at room temperature results in either the same form (α-OADH) with a significant geometric effect on the hydrogen bond length, or a different polymorphic form (β-OADH) [4,31]. Due to the similarity of the protic structures of ADAD and OADH, the X-ray crystal structure determination of deuterated ADAD was performed at 20 °C, −1 °C, and −123 °C, to investigate the structural changes upon deuteration. 

The purposes of the quantum calculations employed in the present study are for: (i) the validation of the structural model and (ii) support for the assignment of the infrared spectra. Regarding (i), the validation is performed both ways: The (presumed) good agreement of the optimized structure with the experimental one justifies the use of the selected computational model for the subsequent evaluation of proton dynamics (i.e., interpretation of the spectra). At the same time, the calculation, in part, confirms the validity of the diffraction-determined structure, and improves upon the structural model, in that it provides a precise and reliable location of the hydrogen-bonded protons. This is often a challenging task for X-ray diffraction techniques, particularly in the case of short hydrogen bonds. The choice of experimental data as starting point is evident, and this has been conducted routinely in a large number of studies. It has to be noted that, while a good match with the experimental data is not a rarity, it is also not self-evident.

The scope of the present work is to elucidate in fine detail the structure and vibrational dynamics of ADAD, thereby improving upon our understanding of its short hydrogen bond, through the use of established experimental techniques, as supported by quantum mechanical (QM) calculations. 

## 2. Results

*Crystal structures of protic ADAD (α, β1, and β2) and deuterated ADAD (αD, β1D, and β2D).* We performed single-crystal X-ray structural analyses at 25 °C, −10 °C, and −123 °C, and labelled the corresponding ADAD structures as α, β1, and β2 for the protic form, and αD, β1D, and β2D for the deuterated ADAD. Using X-ray diffraction and DSC analysis (see SI), we found that there are only two polymorphic crystalline forms in the temperature range between 25 °C and −123 °C, α and β, for both the protic and deuterated ADAD. Upon cooling of the protic ADAD, α converts to β at approximately 4.9 °C. The phase transition is enantiotropic and shows hysteresis. Both polymorphs have a monoclinic unit cell, and the same space group, *P*2_1_/*c* no. 14. Details of the crystal data can be found in Appendix A The unit cell parameters of β1 and β2 and their calculated X-ray powder diffraction patterns (Appendix A) are very similar, showing that both low-temperature structures are isostructural. On the other hand, if we compare the unit cell parameters at low temperature with those at room temperature, and also the powder X-ray diffraction (XRD) patterns at room and low temperature, we find significant differences, confirming that α and β are polymorphic modifications. Figure 1 shows an ORTEP drawing of the ADAD and water molecules with the atomic labelling of the asymmetric unit in the α- and β2-structures. This shows that the main difference lies in the thermal ellipsoids, which are smaller in the β2 structure, due to the less intense thermal motion of the atoms at low temperature (β1 is available in Appendix A). In all cases, the content of the asymmetric unit represents one water and half an ADAD molecule. Additionally, the bond lengths and angles given in Appendix A show that the molecular geometries of the structures are very similar at room temperature and at low temperature. They agree with the expected values for the corresponding bond type. The C1-C1^i^ bond lengths of between 1.181(4) and 1.191(3) Å are consistent with the triple bond between the two C(sp1) atoms. The C1-C2 bond lengths in the interval of 1.461(2)–1.465(2) Å are consistent with a C(sp1)-C(sp2) single bond. The C2-O1 bonds in the range from 1.286(3) to 1.299(2) Å are significantly longer than the C2=O2 bonds, which range from 1.204(2) to 1.214(2) Å, indicating a predominantly single bond and a double bond character, respectively. The comparison of these geometrical parameters with those in the structure of anhydrous acetylenedicarboxylic acid [32] is given in Appendix A. The structural change that occurs during the phase transition is reflected in the molecular packing. Figure 2 and Appendix A show the arrangement of molecules in the α- and β2 structures seen along the *b*-axis and along the *a*-axis, respectively. In the α- and β structures, there are two planar centrosymmetric acid molecules and four water molecules in the unit cell. As can be seen in Figure 3, both forms consist of spiral chains, with alternating strong hydrogen bonds donated by the hydroxyl groups of the acid molecules to the water molecules of the same asymmetric unit (marked with *), and weak hydrogen bonds donated by the water molecules to the carbonyl group of the acid molecules symmetrically connected by a twofold screw axis (marked with i). Within these spiral chains, there is an additional stabilizing contact between the neighboring water molecules, with an O3… O3^i^ contact distance above 3 Å, and an O3-H2… O3^i^ angle ~ 125°. According to this, we could say that the water molecule is through the H2 atom, a donor of the weak bifurcated hydrogen bond. Since this hydrogen bond is very weak, it is not drawn in the figures. Such spirals have a symmetry of the twofold screw axis, and they are parallel to the *b*-axis. Since each acid molecule is centrosymmetric and has two carboxyl groups, such spirals are connected by acid molecules, with a layer that is perpendicular to the *B*-face (defined by edges *a* and *c*) and parallel to the *b*-axis. In the structures of both forms, such layers stack along the diagonal of the *B*-face, with a period of three layers. Adjacent layers are connected by weak hydrogen bonds that are donated by water molecules from one layer, and accepted by the carbonyl group of the acid molecule from the adjacent layer, with the symmetry determined by the *c* glide plane (marked as ii). The hydrogen bonding scheme and the connectivity of the molecules within the layers and between the layers is the same for both forms. The main difference between the α and β polymorphic structures lies in the orientation of the molecules within such layers. The angle between the plane of the acid molecules and the plane of the layer is ~60° in α, and ~30° in the β1 and β2 forms.

Consequently, the α layers are higher, making the *B-face* diagonal longer. On the other hand, the acid molecules within the layer in the α-form come closer to each other in the *b*-axis direction (along the screw axis), resulting in a shorter *b*-edge in α-, compared to the β-form. If we multiply the lengthening factor of the *B*-diagonal (12.6 Å/8.9 Å) and the shortening factor of the *b*-edge (3.87 Å/5.48 Å), we obtain a value of 1. Therefore, the unit cell volumes of both polymorphic forms are very similar. During the phase transition, only the mutual orientation of the molecules within the layers changes, while the hydrogen bonds are not broken. The network of hydrogen bonds remains the same. The geometric parameters of the hydrogen bonds can be found in Appendix A. With decreasing temperature, we can observe a slight shortening of the donor–acceptor contact distances (O^…^O), while the symmetry codes of the connected molecules remain the same, in accordance with the same connectivity scheme (* stands for identity, i for the twofold screw axis, and ii for the glide plane).

The same conclusions can be drawn if we compare the three structures of deuterated ADAD (Appendix A and Table 1). We can observe an analogous phase transition upon the cooling of the αD-structure to the β1D- structure, and the isostructurality of the β1D- and β2D-structures. The bond lengths between the non-hydrogen atoms are practically the same (Appendix A). The usual geometric isotope effect is observed; i.e., a slight increase in the contact distances between the oxygen and hydrogen bonds, but no isotopic polymorphism occurring as in oxalic acid dihydrate. Because of the slightly larger O^…^O contact distances between the molecules, the unit cell volumes of the deuterated structures are slightly larger compared to the protic structures (Appendix A). Deuterated ADAD is isostructural with protic ADAD; to be precise, αD- is isostructural with α-; β1D- and β2D- are isostructural with β1- and β2-. We can also conclude that there is no γ form in the temperature range between −123 °C and 20 °C, neither in the protic nor in the deuterated ADAD.

OADH also crystalizes in two polymorph modifications, α and β [4,31,33]. Both are monoclinic structures with a space group 14. As in α- and β-ADAD, the content of the asymmetric unit consists of one water molecule and half an acid molecule. The scheme of molecular packing and hydrogen bonding in OADH is completely analogous to that in ADAD, as shown in Appendix A. The purple unit cell in α-OADH corresponds to the unit cell transformation from *P*2_1_/*n* (in the published structure [6]) to the *P*2_1_/*c* space group. In β-OADH, we need to exchange the *a* and *c* edges of the published structure [6], make a transformation from the space group *P*2_1_/*a* to *P*2_1_/*c*, and move the origin for ½ of the *a*-edge. For the same reason as in ADAD (different thicknesses of the layers due to different orientations of the acid molecules within the layers), the cell *b*-edge is significantly longer and the *B-*diagonal is significantly shorter in β-OADH compared with α-OADH. Despite the same crystal packing motifs of α- and β-OADH, there is an important difference between them (Appendix A); the hydrogen bond donated by the acid molecule in the protic α-OADH [31] is significantly stronger at 2.487(2) Å compared to 2.524(2) Å in the deuterated α-OADH [6], or 2.538(2) Å in the deuterated β-OADH [6]. This is consistent with oxalic acid being a stronger acid than acetylenedicarboxylic acid. The O^…^O contacts of the hydrogen bonds donated by the water molecules correspond to weak hydrogen bonds in all three structures, as is the case with ADAD. Another important difference between OADH and ADAD is that α-OADH does not undergo a phase transition upon cooling, in either the protic or the deuterated form. β-OADH can only be obtained at room temperature in deuterated OADH. Sometimes the deuterated OADH also crystallizes as α-OADH at room temperature, so the “isotopic polymorphism” does not always occur, and we could not find an explanation for this in the literature.

*QM structure optimization.*Table 2 lists the optimized unit cell parameters for all six models of the title system, together with their computed energies. The models have been constructed on the basis of the presently collected crystal structure data of different polymorphs, and/or different temperatures, and/or different isotopomers. Selected internal geometric parameters of ADAD and its hydrogen bonds are listed in Table 1.

The geometry optimization of all six models yields a very good match with the respective experimental structure data, both in terms of unit cell parameters, as well as with regard to the atomic positions and internal structure of ADAD. The largest observed offset of the computed unit cell parameters in all six models is 1.8%, but most of the computed parameters are within a 1% match with the experimental values (Table 2). While for some models, the calculation either underestimates or overestimates the cell side *a* (but always by a very small margin), it tends to uniformly (but just slightly) underestimate *b* and slightly overestimate *c*. As a result, since *c* is by far the longest side of the unit cell, the optimized volumes are, in general, slightly larger than the measured ones, but again, the discrepancies are very small (1.9% at most). 

In a similar fashion to the unit cell, the calculations also reproduce the hydrogen bond geometry very well, and the offsets from the measured values are very small (Table 1). Specifically, the computed O^…^O distance of the short hydrogen bond is typically shorter than the experimental one by no more than 0.02 Å. At the same time, the O^…^O separation in one of the two longer hydrogen bonds (with water acting as a donor and the C=O group of acetylenedicarboxylic acid as an acceptor) is underestimated by approximately 0.05 Å, while in the other, it is overestimated by approximately 0.02 Å. Much larger offsets can be observed for the positions of the hydrogen atoms—typically, the calculations predict at least 0.1 Å longer O–H distances than what is measured–but this is evidently due to the limited ability of XRD to resolve the location of the hydrogen atoms, due to the fact that the electron density around the hydrogen atoms is very low. 

The selected bond lengths within the acetylenedicarboxylic acid molecule are also fairly well reproduced, according to calculations (Table 1). Of those, the carbonyl (C=O) bond length features the weakest (yet still acceptable) match; the computed values were 0.03–0.05 Å longer than measured, while the C–O bond length of the carboxylic group was overestimated by 0.01–0.03 Å. The computed lengths of the C≡C bond are longer than their experimental counterparts by 0.02–0.03 Å, whereas the calculations underestimated the C–C bond length by 0.01–0.02 Å.

Finally, the superposition of a computed crystal structure with its experimental counterpart features an RMSD of approximately 0.06 Å per atom (with the hydrogen atoms excluded for the reason given above) for any of the six presently discussed structures. As it has been postulated that an RMSD value of up to 0.25 Å still represents an acceptable match [34], it can be concluded that the present calculations deliver an excellent agreement with the experimental crystal structure. 

In any case, all of the six optimized models feature remarkably similar bond lengths, hydrogen bond geometries, and unit cell parameters when compared to their experimental counterparts. The similarity is further confirmed by the computed energies (Table 2)—the energy difference between the models is, in most cases, marginal, and the largest difference of 0.29 kcal/mol per stoichiometric unit suggests that the models differ, at worst, by the value of thermal energy (k_B_T), even at −123 °C, meaning that unlike the computed arrangement of molecules in the crystal structure that confirm clear differences between the α- and β- polymorphs—the computed energy differences are thermodynamically insignificant. Although this information probably cannot discern the subtle details governing the phase transition between the polymorphs, it is encouraging that all of the herein reported calculations of the structure are a very good match with the experimental observations. 

*Infrared measurements and QM frequency calculations.* The attenuated total reflection (ATR) spectra of the protiated and deuterated ADAD recorded at T = 25 °C and T = −150 °C are shown in Figure 4 and Figure 5. The overall assignment shown in Table 3 and Table 4 was verified using the calculated frequencies shown in Table 5, and by the assignment of oxalic acid dihydrate [8]. As we are interested in the correlations between the hydrogen bonding properties reflected in the vibrational spectra, we will mainly concentrate on the vibrational modes that are associated with hydrogen bonding. The assignment of the corresponding bands are with assistance from the computed normal modes and harmonic frequencies (see Appendix A).

The high-frequency region begins with the symmetric and antisymmetric OH stretching vibration of water molecules. The splitting of the stretchings at 3508 cm^−1^ and 3382 cm^−1^ exceeds the usual splitting found in the spectra of carboxylic acid hydrates (~50 cm^−1^). Cooling it to −150 °C increases it from 127 cm^−1^ to 148 cm^−1^, but deuteration reduces it to 106 cm^−1^ and 115 cm^−1^. While the symmetric vibrations are fairly well reproduced according to QM calculations (computed as 3360 cm^−1^–3390 cm^−1^), the frequency of the antisymmetric modes are apparently overestimated, and are computed to 3630 cm^−1^–3640 cm^−1^. This similarly holds for the computed frequencies of the deuterated system. Accordingly, the splitting of the water OH stretching modes is overestimated according to calculations, amounting for approximately 265 cm^−1^ for the protiated, and 195 cm^−1^ for the deuterated species, respectively.

In the experimental spectrum, the water stretches are followed by a ~1000 cm^−1^ wide hump, topped by six subpeaks. The most intense sub-peak, at 25 °C, is the peak at 2497 cm^−1^. When cooled, it gains intensity and shifts to higher frequencies. In addition, the shoulder on the red flank of this hump develops during the cooling of the sixth sub-peak, at 2288 cm^−1^. The hump and its sub-structure are completely removed via deuteration. Therefore, it is obvious that the hump, which is centered at ~2500 cm^−1^, is a part of the wide OH stretching. It is difficult to exactly determine the range that expands, and the maximum. The intensity of the hump decreases with cooling, mostly at its red portion. The hump is separated by a depression at ~2139 cm^−1^ from a broad band-like feature (BLF) at 1972 cm^−1^. The irregular contour, and especially the sharp topping peak at 1981 cm^−1^, indicate a composite origin for the BLF, which is even more pronounced in the cooled sample. The cooling leads to an increase in absorption by the red flank of the BLF, which causes a redshift of ~50 cm^−1^. The BLF found in ADAD is not unique. A similar pattern is found in the spectra of oxalic acid dihydrate [7,8,14] and perfluorated fatty acid monohydrate [17]. Moreover, the BLFs in the spectra of the aforementioned compounds are also a complex structure of at least two components, which are similarly sensitive to temperature and/or deuterium exchange, as in the case of ADAD.

The calculations predict the corresponding OH stretching frequencies to be in the range between 2260 and 2470 cm^−1^. Qualitatively, the agreement is satisfactory, considering several limitations of the approach (harmonic approximation, no information on the intensities, and factors contributing to band broadening). The fact that the computed frequencies tend to be slightly blueshifted with respect to the assumed centroid of the experimental OH stretching band is in line with the slightly overestimated shortness of the corresponding hydrogen bond, for which the O^…^O distance is predicted to be approximately 0.02 Å shorter than that measured via XRD (Table 1). Nevertheless, a more quantitative assessment of the factors contributing to the shaping of this band, together with the observed temperature effects would require a much more complex treatment, which is beyond the scope of the present work. It should also be noted that a precise characterization of the experimental band is difficult due to its complex shape, both in the case of the protiated and the deuterated form. 

The strong band, with a peak at 1647 cm^−1^ is clearly assigned to the C=O stretch, and it shifts by 7 cm^−1^ when cooled, shifts only by 1 cm^−1^ when deuterated, and shifts by 12 cm^−1^ when the deuterated sample is cooled. This assignment is neatly reproduced by the computed modes at 1630–1640 cm^−1^, which are composed of the C=O stretching motion, but they also feature a noticeable share of in-plane CO-H bending. A very small frequency shift upon deuteration implies weak coupling between the C=O stretch and water bending, which is, in general, present in such systems. Traces of water bending also appear in the aforementioned computed modes. The C=O stretch of the protiated sample has a shoulder on its red flank at 1603 cm^−1^, which separates upon cooling, to 1597 cm^−1^. Another small peak at 1523 cm^−1^ is present at the base of the band, in both the protiated and deuterated samples. A weak third sub-peak is present only in the cooled deuterated samples. Note the widening of the lower part of the C=O band base that occurs during deuteration. 

An unusual event appears upon deuteration with the band at 1435 cm^−1^. It is removed, and instead of absorption, a sharp dip (1423 cm^−1^) occurs. Its shape suggests that it is the result of an Evans process [35,36]. Namely, the hump of the high-frequency part (OH stretch) has been shifted to ~1400 cm^−1^ via deuteration, and the mode (OD stretch) has the ability to form an Evans transmission with the corresponding mode from the low-frequency region of appropriate symmetry. In contrast to that, the computed modes with a noticeable share of OH stretching motion appear at higher frequencies, namely at 1680–1810 cm^−1^ (virtually decoupled from other vibrations), and at 1530cm^−1^–1590 cm^−1^ (coupled with C=O stretching). Another related discrepancy between the calculation and the experiment is in that the C=O stretching band of the deuterated isotopomer is located at 1634 cm^−1^, which is significantly higher than according to computational prediction. Both discrepancies may be attributed to the peculiar coupling of the two involved modes found using calculations, but it should be noted that locating the centroid of the broad OD (OH) stretching mode in the experimental spectrum represents a challenging task.

The band at 1437 cm^−1^ in the protiated spectrum is assigned to the in-plane bending of the CO-H coupled with the C-OH stretching, as is characteristic of hydrogen bond carboxylic acids [24]. The latter mode is represented by a strong split peak, at 1283 cm^−1^ and 1286 cm^−1^. Both modes, particularly the one with a higher frequency, are reasonably reproduced using calculations (1450 cm^−1^–1470 and 1270 cm^−1^–1340 cm^−1^), but the latter appears to also include a sizable contribution of in-plane CO-H bending. This assignment is supported by the deuteration shift of the band at 1437 cm^−1^ to 1091 cm^−1^, and the split band at 1361 cm^−1^ and 1305 cm^−1^. 

The redshift of the frequency during deuteration is also an indication of out-of-plane CO-H bending. In the protiated spectrum, a redshift is found at 1071 cm^−1^. This band shifts to 786 cm^−1^ when deuterated. Cooling affects the frequencies of both bands via blue-shifting for 36 cm^−1^ and 35 cm^−1^, respectively. These displacements are relatively large compared to the displacements of the corresponding in-plane bending (Table 3 and Table 4). For both isotopomers, the calculation significantly overestimates the frequency of this mode, predicting it at 1200 cm^−1^–1210 cm^−1^ for the protonated and at 870 cm^−1^–880 cm^−1^ for the deuterated species. While this discrepancy is hardly attributed to the just slightly underestimated O^…^O distance alone, the relative error for this mode appears to be roughly the same for both isotopomers. 

*Raman measurements.* As expected, the Raman spectra shown in Figure 6 and Figure 7 differ significantly from the infrared spectra. The most prominent band belongs to the stretching oscillation of the C≡C group, located at 2256 cm^−1^ (Table 6 and Table 7). The computed frequencies of approximately 2260 cm^−1^ are in excellent agreement with this. The C≡C stretching band moved to a higher wavenumber, up to 2374 cm^−1^, when the temperature was lowered; this feature could not be reproduced by the simple calculations. Similar trends were observed in the deuterated form when the temperature was lowered, although the peak frequencies were much lower (2243 cm^−1^ at RT and 2255 cm^−1^ at −110 °C). An antisymmetric broad band centered at 3402 cm^−1^, and a shoulder at 3470 cm^−1^ corresponds to the water OH stretching vibrations of Ag and Bg symmetry (Figure 6). Both bands are present in the spectra recorded at room and at low temperature. The fingerprint area begins with the complex structure of the C=O stretching (Figure 8). Two components can be localized at room temperature, while the third component is visible without band decomposition only at low temperature. The distinguishability of the bands is reversed via deuteration. Only two components are visible at low temperature, and three at RT. The change in the frequency of the C=O bands during deuteration exhibited coupling between the C=O stretch and the H_2_O bending modes, as well as with the C-O-H(D) moiety. The complexity of the C=O stretching mode is also confirmed via calculations; however, the coupling scheme for this mode appears to be slightly different from experimental assignments, as discussed above. 

## 3. Discussion

In the introduction, it was briefly mentioned that the ABC type of the IR spectra of compounds with strong hydrogen bonds is the most frequently used characteristic of the fine structure occurring in the literature. In the following presentation of the experimental results, however, we will introduce and comment on the spectra of a system consisting of only two features that form the debatable part of the OH stretch. Many examples appear in the literature, and they are then either marked as an AB type of vibration spectrum, or simply neglected. The presentation of the spectra will show that it is the band C with the lowest frequency of the ABC type that has the characteristics of band B, when only AB bands are easily observed. To avoid misinterpretation, we will therefore use C1 for the high-frequency part (3250 cm^−1^–2150 cm^−1^), and a C2 label for the low-frequency part of the spectra (2150 cm^−1^–1750 cm^−1^). Both regions are marked in Figure 4.

The molecular and crystal structures of ADAD was first published in 1947 [2]. As far as the hydrogen bond between the carboxyl groups and the water molecule is concerned, it is almost the same as the bond presented in this publication (2.56 Å). However, in the previous study, it was not possible to distinguish between the neutral and the ionized carboxyl groups. The latest work clearly shows that there is no proton transfer. This is also in accordance with the vibration spectra and the theoretical results. 

X-ray diffraction studies have shown two polymorphic formations: α above 4.9 °C and β below 4.9 °C. The IR spectra are only marginally affected by the cooling. The most important changes that occur during cooling concern the complex component C2. It consists of an absorption intensity increase that mainly affects its red part. The topping peaks of C1 segment are blue-shifted by 10 cm^−1^–20 cm^−1^. A slight redshift (7 cm^−1^) occurs at the carbonyl stretch, and even less (3 cm^−1^) at the band at 1435 cm^−1^, which is associated with OH in-plane bending. This band is of constant intensity. Most interesting is the effect of cooling, which occurs in the band at 1071 cm^−1^. It is shifted with a much higher intensity, to 1107 cm^−1^. Both band changes are characteristic for bending outside of the OH plane. Unfortunately, the present calculations are generally unable to elucidate the temperature- and structure-dependent spectral features in such fine detail.

The results of the deuteration are also important. In the IR spectrum of the isotopomer, the C1 segment lacks the characteristic series of substructure peaks, but the C2 component of the complex OH feature, which occurs at ~1900 cm^−1^, shows small changes compared to the spectrum of the protiated system. This change consists of a slight intensification of the red side of the feature, and thus, of a redshift of its center by 12 cm^−1^. 

The contour of the deuterated C2 component also differs. Cooling affects the deuterated C2 feature, resulting in a redshift of approximately 40 cm^−1^. The depression that separates the C2 component from the C=O stretch disappears almost completely. Important for the discussion, the approximate origin of the components of the complex OH line are the bands at 1443 cm^−1^ and 1257 cm^−1^, which are assigned to the complex modes of bending and C-O stretching, which mainly take place in the OH plane. Deuteration shifts these bands to 1083 cm^−1^ and 1353 cm^−1^, respectively. This corresponds to the spectra of the carboxylic acid dimers, although the hydrogen bond strength in the latter is significantly weaker. 

The broad OD stretching range is difficult to define in the IR spectra. The high-frequency wing obviously starts under the water stretching bands and expands toward C=O stretching, but its maximum and the low-frequency part are obscured by the internal modes of the acetylenedicarboxylic acid, including the modes of its OH group. The interpretation shows only the narrowing of the high-frequency segment, but the appearance of the Evans dips at 1480 cm^−1^ and 1000 cm^−1^ indicates that the OD strain goes even further to lower wavenumbers. The calculations give ~2200 cm^−1^ for the mainly acidic OH route, but this seems to be too high. 

*Comparison with the vibrational spectra of oxalic acid dihydrate*. The absence of environmental influences on the bands, which are characteristic for the hydrogen-bonded groups, offers a unique opportunity to study the influence of the bond length on these bands. The shortest O…O distance in oxalic acid dihydrate (2.49 Å) is slightly shorter than the corresponding one in acetylenedicarboxylic acid dihydrate (2.57 Å). It was expected that the most pronounced differences in OH band frequency and shape should be observed. However, as shown in Figure 9, both the shape and the position of the toppings are almost identical. This is especially true for the deuterated species (Figure 10). The only exception is the band component at 1825 cm^−1^, which is only found in the spectrum of oxalic acid dihydrate. The similarities in the shapes and frequencies of the broad OH stretching with complete fine structure indicates that similar mechanisms are responsible for the band shape. The appearance of the components (AB or ABC) can be satisfactorily explained via the presence of Fermi resonance between the broad OH stretching, and the in-plane, out-of-plane OH bending and the combination of these bands [19]. However, the fine structure of the OH stretching, characterized by several additional band minima and maxima, still requires adequate explanation. A very similar fine structure has been observed in the spectra of the cyclic acid dimers [19,23,37,38]. The theoretical explanation of the complex band shapes of the broad OH stretchings is based on the coupling of the OH stretching with the intermolecular stretching of the hydrogen bonds and the occurrence of the Davydov coupling between the OH stretchings in the dimer. Although the proposed theory has been developed for isolated acid dimers, it has been shown that it can be successfully applied, not only to various polycrystalline acid dimers [39], but also to crystals with hydrogen-bonded molecular chains [40]. The non-existence of the dimer structure, and thus, the proximity of the two stretching modes, can, in our case, be replaced by the stacking of the hydrogen bonds between the carboxylic OH and the water molecules in both crystal hydrates, which allows for efficient coupling between the OH stretching modes. The delocalization of the OH stretches leads to the formation of exciton oscillations and the Davydov coupling of adjacent OH stretching modes. Therefore, this theoretical approach is also suitable for explaining the fine-structured OH stretching in the infrared spectra of acetylenedicarboxylic acid dihydrate and oxalic acid dihydrate. Furthermore, the increased anharmonicity of the potential energy surface leads to the coupling of different degrees of freedom of vibration [23]. Especially, the OH stretching mode can be coupled to low-frequency modes, which changes the length of the hydrogen bond. Such a coupling leads to the formation of sidebands that are shifted with respect to the pure OH stretching excitation. 

Among the rare differences are those found in both the spectra of hydrates belonging to the “in plane” and “out of plane” positions of the OH deformation modes. The extension of the hydrogen bond length shifts these bands by ~10 cm^−1^ to higher frequencies. Further differences that are presented in Figure 9 and Figure 10 are due to different internal modes of acetylenedicarboxylic acid and oxalic acid.

The differences can also be seen in the Raman spectra of the two components (Figure 11). The broad absorption between the 2200 cm^−1^ and 1200 cm^−1^ associated with the OH stretching vibration is only present in the spectrum of oxalic acid dihydrate. In this region, only a broad band with a low intensity can be observed in the Raman ADAD spectrum. This OH stretching type of vibration should be forbidden (A_g_ type) in the Raman spectra of both compounds, but the partial symmetry break (a loss of the inversion center) can increase the intensity of this mode [41]. Moreover, the origin of the broad bands positioned in the Raman spectrum of oxalic acid dihydrate at 1600 cm^−1^ is attributed to the proton polaron, a mode that corresponds to the vibration of the defects in the form of H_3_O^+^.COOHCOO^−^. This type of defect cannot be observed in the Raman spectrum of ADAD. 

The presence of the polaron type of vibration in the Raman spectra may explain the differences in the infrared spectra. The band at 1825 cm^−1^, present only in the spectrum of oxalic acid dihydrate, can be thus attributed to the presence of the polaron vibration. The deuterated infrared spectrum confirms such an assignment. The band at 1825 cm^−1^, due to crystal defects, moves to lower frequencies and becomes undetectable, due to strong overlapping with the intrinsic acid bands. Thus, it becomes almost identical to the region in the front of carboxylate stretching, in the spectra of the deuterated ADAD and of oxalic acid dihydrate.

## 4. Material and Methods

ADAD (95%) was purchased from Sigma-Aldrich. Well-formed crystals in the form of dihydrates were obtained from water solution as colorless plates. The deuterated form was grown from D_2_O solution with double recrystallization. Deuterated crystals were of the same color and similar shape as the protic crystals. Large and thick crystals can grow with time for both cases.

*Vibrational spectroscopy*. The infrared spectra were recorded with a Bruker Vertex 80 spectrometer using a diamond ATR cell (Specac, Golden Gate, Orpington, UK) with KRS-5 optics. Typically, 128 spectra were added at a nominal resolution of 2 cm^−1^. Low temperature measurements were performed with the same ATR cell, using the low temperature ATR base plate. To maintain the optimal signal-to-noise ratio at a different temperature before each measurement, a fresh background single-beam spectrum was measured at the same temperature as the sample. The recorded ATR spectra were corrected for frequency-dependent penetration depth using the OPUS software method. Fourier transform (FT)-Raman spectra were recorded with a Bruker Ram II spectrometer, using a 1064 nm laser. Typically, 256 scans were averaged, with a nominal resolution of 4 cm^−1^. Low temperature measurements were performed with a self-made low-temperature cell. The crystals were sealed in 5 mm NMR tubes and tempered by the vapor of liquid nitrogen. The temperature was controlled via a Digisense temperature controller. Additional RT and low-temperature Raman spectra were obtained via a HORIBA Jobin Yvon T64000 Raman spectrometer operating in micro single and micro triple mode, with a 532 nm semiconductor laser DPSS from Changchun New Industries Optoelectronics Tech, using a nitrogen-cooled Linkam cell.

*X-ray diffraction.* Single-crystal diffraction data of protic and deuterated ADAD were collected on an Agilent SuperNova dual-source diffractometer with an Atlas detector at 20(1), −10(1), and −123(1) °C. Crystal structures of protic ADAD are labelled α, β1, and β2 at 20, −10, and −123 °C, and of deuterated ADAD, αD, β1D, and β2D, respectively. The data were processed using CrysAlis PRO software (version 1.171.39.46e, Rigaku Oxford Diffraction, Yarnton, UK) [42]. The structure was, in all cases, solved using direct methods via SIR97 [43]. A full-matrix, least-squares refinement on F^2^ was employed with anisotropic displacement parameters for the non-hydrogen atoms. H atoms in all structures were located in difference Fourier maps. Their positional parameters were refined, together with their isotropic displacement parameter. SHELXL2018/3 software (Goettingen, Germany, 2018) [44] was used for the final structure refinement and interpretation. The structures were drawn using ORTEPIII [45] and Mercury [46]. Details regarding the crystal data, data collection, and structure refinement are given in Appendix A. All crystallographic details for the structures α, β1, and β2, and αD, β1D, and β2D, have also been deposited with the Cambridge Crystallographic Data Centre as supplementary publication numbers CCDC 2160307, 2160309, 2160306,1 2160304, 2160305, and 2160308, respectively. These data can be obtained free of charge via www.ccdc.cam.ac.uk/conts/retrieving.html, accessed on 25 May 2022 (or from the CCDC, 12 Union Road, Cambridge CB2 1EZ, UK; fax: +44-1223-336033; e-mail: deposit@ccdc.cam.ac.uk).

*The X-ray powder diffraction pattern* of protic ADAD at 20(1) °C was collected using a PANalytical X’Pert PRO MPD diffractometer with reflection geometry and a primary side Johansson type monochromator, with CuKα1 wavelength (1.540596 Å). 

*The DSC curve* of protic ADAD was measured using a Mettler Toledo TGA/DSC1 thermo analyzer. A total of 3.197 mg of the sample was weighed into an aluminum crucible and covered with an aluminum lid. An empty pot was used as a reference. The measurement was performed according to the following temperature program in the air flow: cooling from 25 to −70 °C at a speed of 10 °C/min and reheating from −70 to 25 °C at a speed of 10.0 °C/min. The air flow was 50.0 mL/min. The average enthalpy of the phase transition was 7.4 J/g (See Appendix A).

*QM Calculations.* All herein reported crystal structures (α-polymorph, measured at 20 °C, and β-polymorph, measured at −10 and −123 °C, in both the deuterated and protic forms; six distinct structures altogether) were treated using the program package VASP v. 5.3.5. [47,48,49,50]. Each of the structures studied via X-ray diffraction analysis served as a starting point for a full geometry optimization, including the atomic positions and the unit cell parameters, but with the preservation of the *P*2_1_/*c* crystal structure symmetry. Since the calculation of the electronic structure is insensitive to the atomic masses due to the Born–Oppenheimer approximation, these optimizations are mass-independent. Following geometry optimization, the harmonic frequencies and normal modes were computed for all six models using the respective masses of the H and D isotopes. All QM calculations were conducted in the Density Functional Theory (DFT) formalism under periodic boundary conditions. A revised version [51] of the Perdew-Burke-Ernzerhof (PBE) functional [52], corrected for dispersion effects using the DFT-D3 method of Grimme [53], was employed together with the Projector Augmented Wave ultrasoft pseudopotentials [54,55] and a plane wave basis set with a cutoff of 500 eV. Electronic integrals in the reciprocal space were computed on a 4 × 6 × 3 Monkhorst-Pack mesh of *k*-points [56]. A threshold of 10^−6^ eV was set as the convergence criterion for the electronic structure relaxation, whereas for geometry optimization, a force-stopping criterion of 0.003 eV/Å was used. The assignation of normal modes was assisted via JMol visualization software [57], whereas optimizations were visualized using the VMD v. 1.9.3 program [58].

*Potential energy distribution* (*PED) calculation.* To facilitate the assignment of the observed vibrational bands in the crystal, normal coordinate analysis for the trans conformer of acetylenedicarboxylic acid was performed using Gaussian09 (Revision A.02, Gaussian, Inc., Wallingford, CT, USA) [59]. The geometry was constrained to a **C_2h_** point group (*trans* conformer), which resulted in the appearance of one imaginary frequency, indicating this was not the most stable conformation of the molecule in a gaseous state. PED among the normal modes was calculated using the BALGA program [60,61].

## 5. Conclusions

ADAD represents a complex with a strong hydrogen bond between the carboxylic OH and a water molecule, and thus it shows a typical infrared spectrum with a broad and complex OH stretching vibration. An X-ray re-examination of the ADAD crystal structure confirms the O^…^O distance of the short hydrogen bonds, and clearly demonstrates the differences in the bond lengths of both oxygen atoms with respect to the carbon atom in the carboxylic moiety, which implies the neutral structure of the complex. A neutral structure was also confirmed via vibrational spectroscopy, since no proton transfer was observed. In the temperature range between 25 and −23 °C, there are only two polymorphic crystalline forms, α and β, for both the protic and deuterated ADAD. Upon the deuteration of ADAD, only the usual geometric isotope effect is observed. Deuterated α is isostructural with protic α, and deuterated β with the protic β structure, respectively. Neither calorimetric nor X-ray diffraction measurements support the third, γ form.

The herein employed quantum calculations are in excellent agreement with the experimental diffraction structure, thereby validating the structural model that was used for the evaluation of the proton dynamics. Although remaining on the level of harmonic approximation, calculations have been able to enhance the interpretation of the infrared spectra, fairly reproducing most of the bands that are related to the hydrogen bond, whereas the discrepancy between the computed and observed OH stretching band is understandable for a hydrogen bond of such shortness.

The comparison of the structure and vibrational spectra between ADAD and oxalic acid dihydrate reveals some interesting details. The crystal structures of both crystal hydrates are almost the same, with only the O^…^O distances of the strongest hydrogen bonds differing by 0.08 Å. It was expected that a larger O^…^O distance in the ADAD crystal would significantly alter the infrared and Raman spectra, especially in the frequency and shape of the acidic OH stretching vibration. However, both the shape and frequency are almost identical, with all subpeaks topped on the broad OH stretch. The O^…^O distance dependence is only applicable for the in- and out-of-plane deformation OH modes. 

The origin of the broad OH band shape was rationalized in a similar way to the acid dimers: the anharmonicity of the hydrogen bond potential gains coupling of the OH stretch with the low frequency hydrogen bond stretchings. The commutation of the OH stretch in two humps, assigned as C1 and C2, is caused by Fermi resonance. The fine structure, similar to that found in the spectra of acid dimers, is attributed to Davydov coupling.

The presence of polarons due to the H_3_O^+^.COOHCOO^-^ defect was not observed in the vibrational spectra of ADAD. Hence, the OH stretching broadening cannot be influenced by the polaron mechanism, as introduced by Fischer et al. [20]. On the contrary, this effect can alter the shape of the broad OH stretching in the infrared spectrum of oxalic acid dihydrate. The band positioned at 1825 cm^−1^ in the spectrum of oxalic acid dihydrate can be assigned to polaron vibration.

## Figures and Tables

**Figure 1 ijms-23-06164-f001:**
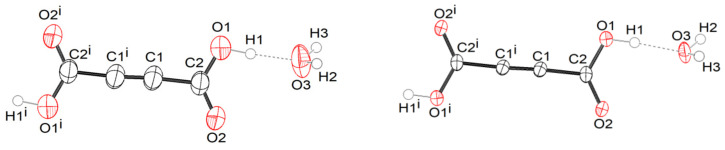
Atom labelling scheme of water and the ADAD molecules in the α (**left**) and β2 (**right**) structures. The displacement ellipsoids are at a 50% probability level (i: –x, –y, –z+2).

**Figure 2 ijms-23-06164-f002:**
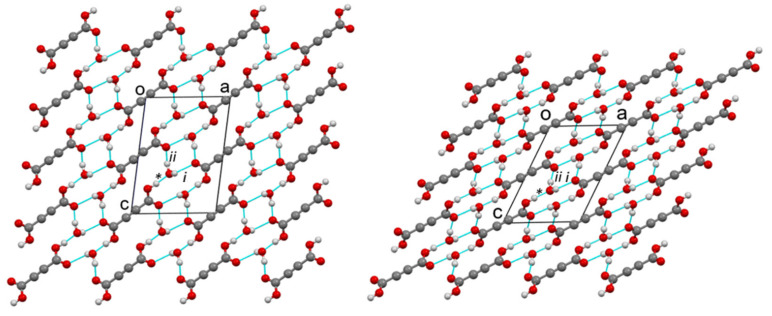
Molecular packing of the α- (**left**) and β2- (**right**) structures, viewed along the *b*-axis. Blue lines represent hydrogen bonds with symmetry codes **: x,y,z;* i: 1−x,1/2+y,3/2−z; ii: x,−1/2−y,−1/2+z.

**Figure 3 ijms-23-06164-f003:**
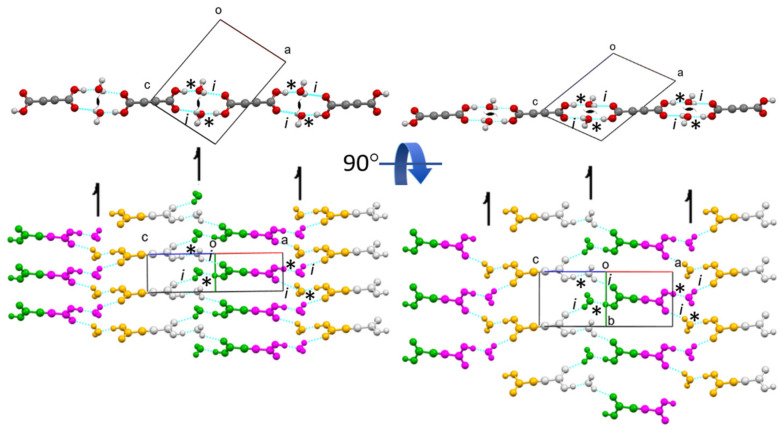
Layers that build ADAD in both polymorphic forms: α (**left**) and β2 (**right**). The upper drawings represent layers as viewed along *b*-axis. The bottom drawings show layers in the plane of the paper. Blue lines represent hydrogen bonds. * represents hydrogen bond connection of molecules in the same asymmetric unit, and I represents *a* symmetry relation by twofold screw axis.

**Figure 4 ijms-23-06164-f004:**
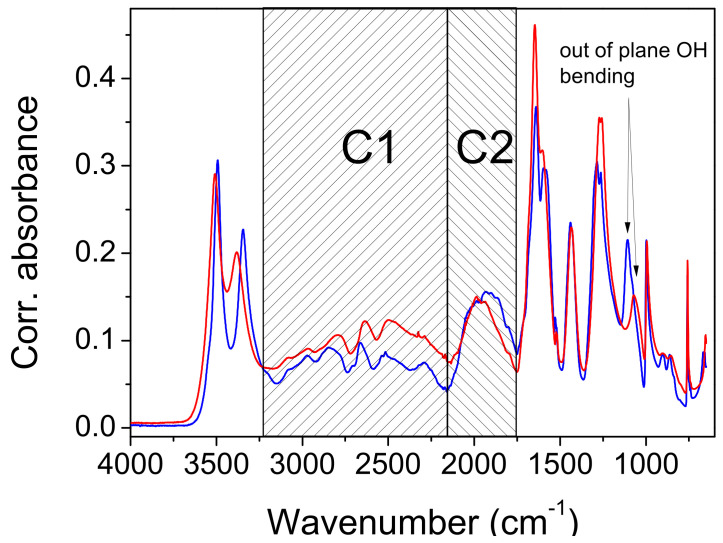
The infrared spectrum of ADAD recorded at T = 25 °C (red line) and T = −150 °C (blue line). C1 and C2 represent high- and low-frequency regions of the broad OH stretch, respectively.

**Figure 5 ijms-23-06164-f005:**
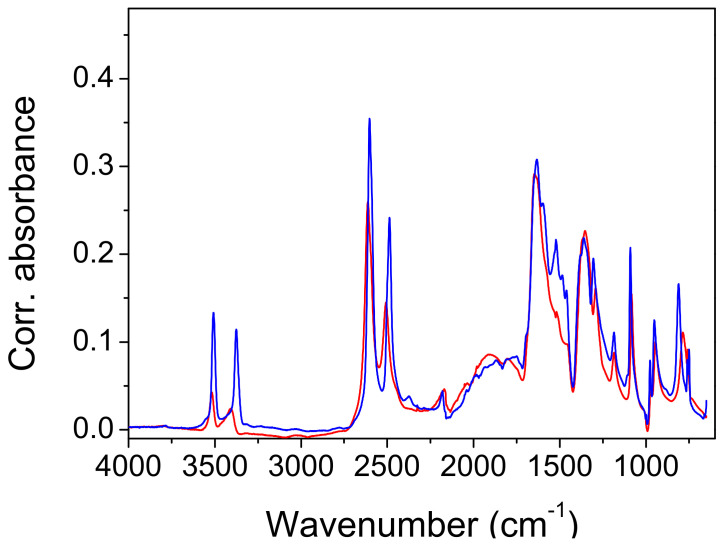
The infrared spectrum of ADAD (deuterated) recorded at T = 25 °C (red line) and T = −150 °C (blue line).

**Figure 6 ijms-23-06164-f006:**
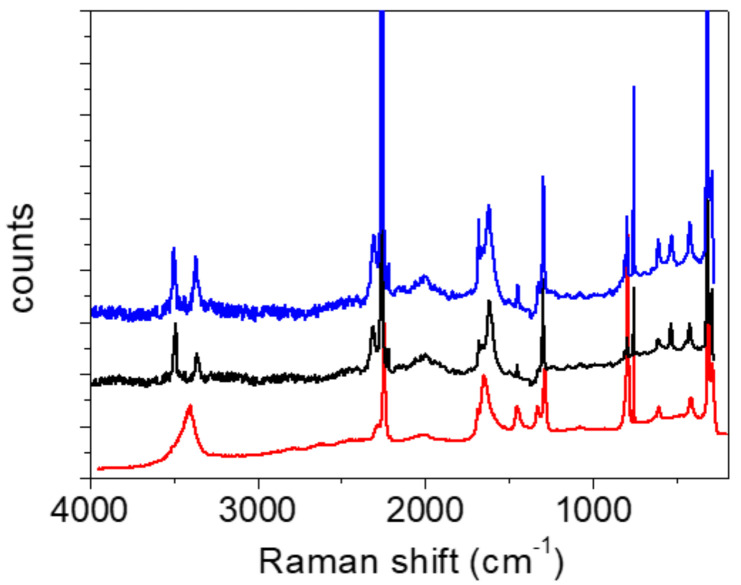
The Raman spectrum of ADAD, recorded at T = 25 °C (red line), T = −130 °C (black line), and T = −160 °C (blue line). Spectra were recorded with 585 nm laser.

**Figure 7 ijms-23-06164-f007:**
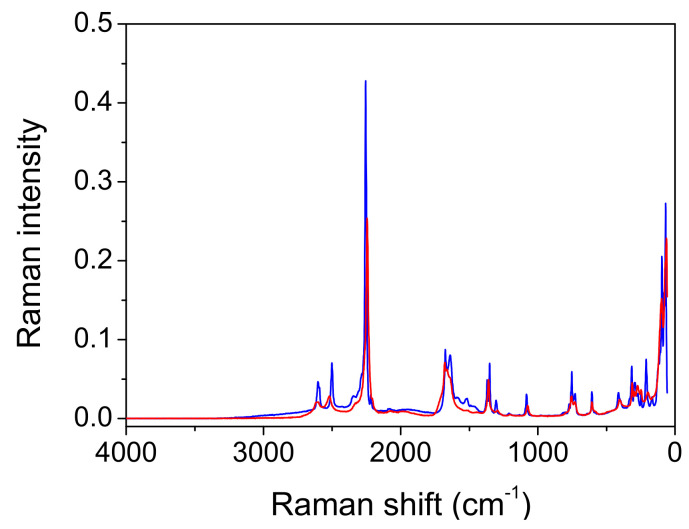
FT-Raman spectrum of ADAD (deuterated), recorded at T = 25 °C (red line) and T = −110 °C (blue line). FT-Raman spectrum of ADAD (protic) is presented in SI (Appendix A).

**Figure 8 ijms-23-06164-f008:**
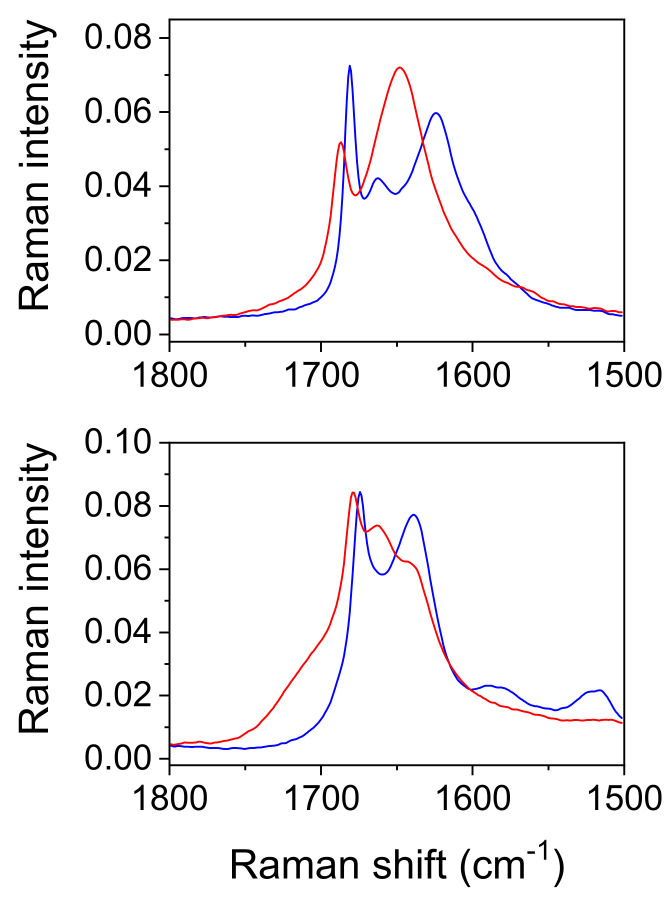
Enlarged region of carboxylate stretching vibration in FT Raman spectra of ADAD. Red spectra are recorded at room temperature, and blue at −110 °C. (**Upper graph**): protiated ADAD, (**bottom graph**): deuterated ADAD; spectra are scaled to same intensity.

**Figure 9 ijms-23-06164-f009:**
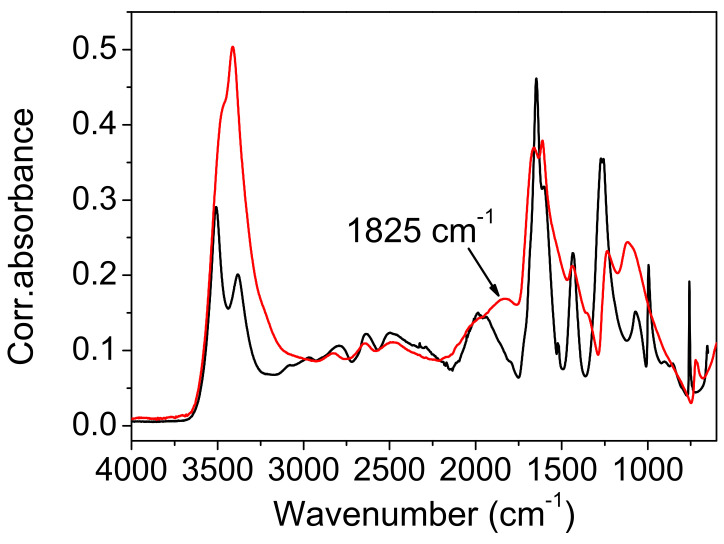
ATR spectra of oxalic acid dihydrate (red line) and acetylenedicarboxylic acid dihydrate (black line).

**Figure 10 ijms-23-06164-f010:**
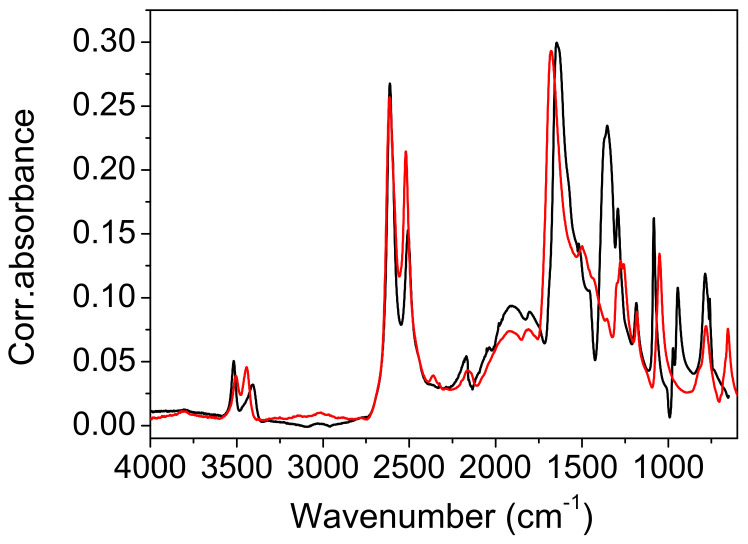
ATR spectra of deuterated oxalic acid dihydrate (red line) and deuterated acetylenedicarboxylic acid dihydrate (black line).

**Figure 11 ijms-23-06164-f011:**
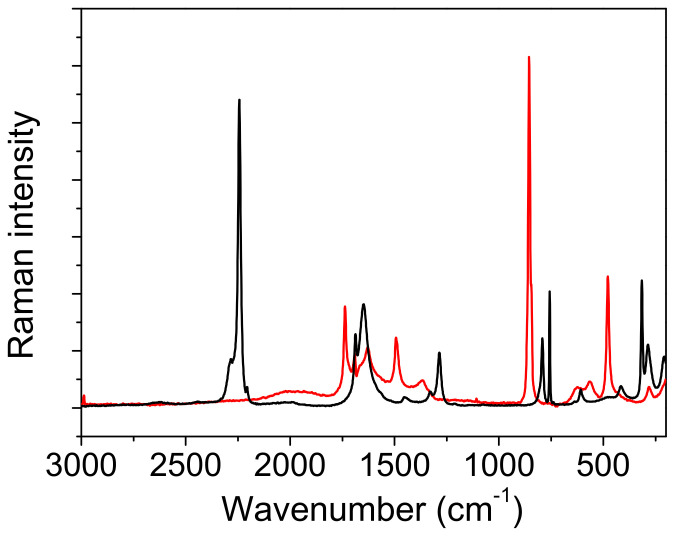
FT Raman spectra of oxalic acid dihydrate (red line) and acetylenedicarboxylic acid dihydrate (black line).

**Table 1 ijms-23-06164-t001:** Selected interatomic distances of the optimized structures listed in Table 2. The corresponding experimental values are added in parentheses.

°C	r(O^…^O)_1_ [Å]	r(O—H)_1_ [Å]	r(C–O) [Å]	r(C=O) [Å]	r(C–C) [sÅ]	r(C≡C) [Å]	r(O^…^O)_2_ [Å]	r(O^…^O)_3_ [Å]
α, H 20	2.520	1.052	1.316	1.254	1.451	1.213	2.744	2.929
	(*2.542*)	(*0.952*)	(*1.285*)	(*1.204*)	(*1.465*)	(*1.181*)	(*2.724*)	(*2.904*)
α, D 20	2.523	1.053	1.316	1.257	1.453	1.214	2.742	2.930
	(*2.552*)	(*0.932*)	(*1.287*)	(*1.205*)	(*1.463*)	(*1.183*)	(*2.782*)	(*2.909*)
β, H −10	2.519	1.052	1.311	1.254	1.448	1.211	2.728	2.886
	(*2.539*)	(*0.921*)	(*1.289*)	(*1.209*)	(*1.462*)	(*1.185*)	(*2.772*)	(*2.867*)
β, D −10	2.527	1.054	1.313	1.257	1.453	1.214	2.731	2.901
	(*2.552*)	(*0.939*)	(*1.294*)	(*1.209*)	(*1.461*)	(*1.189*)	(*2.782*)	(*2.882*)
β, H −123	2.517	1.051	1.307	1.247	1.447	1.210	2.715	2.885
	(*2.533*)	(*0.915*)	(*1.299*)	(*1.214*)	(*1.466*)	(*1.189*)	(*2.765*)	(*2.851*)
β, D −123	2.521	1.053	1.306	1.248	1.449	1.212	2.712	2.888
	(*2.542*)	(*0.890*)	(*1.295*)	(*1.211*)	(*1.465*)	(*1.191*)	(*2.770*)	(*2.860*)

Note that the three O^…^O distances pertain to three distinct hydrogen bonds, of which the short hydrogen bond with the ADAD molecule as donor and water as acceptor is labelled as ()_1_, whereas the two moderately long hydrogen bonds with water as donor and the carbonyl group of ADAD as acceptor are labelled ()_2_ and ()_3_.

**Table 2 ijms-23-06164-t002:** DFT-optimized unit cell parameters (*a, b, c*) and unit cell volumes (*V*). The starting point of each calculation was one of the herein reported experimental XRD structures (α and β polymorphs) recorded at different temperatures for both undeuterated (H) and deuterated (D) species.

Starting Structure °C	*a* [Å]	Δ*a*/*a* [%]	*b* [Å]	Δ*b*/*b* [%]	*c* [Å]	Δ*c*/*c* [%]	*V* [Å^3^]	Δ*V*/*V* [%]	*E* [eV]	Δ*E* [kcal/mol]
α, H 20	8.000	+0.21	3.797	−1.81	11.107	+0.36	334.03	−1.28	−203.473	0.06
	(*7.983*)		(*3.867*)		(*11.067*)		(*338.35*)			
α, D 20	7.985	−0.15	3.818	−1.37	11.099	+0.33	334.82	−1.25	−203.476	0.02
	(*7.997*)		(*3.871*)		(*11.063*)		(*339.07*)			
β, H −10	7.050	−0.16	5.409	−0.77	9.966	+1.46	341.85	+0.49	−203.470	0.09
	(*7.061*)		(*5.451*)		(*9.823*)		(*340.20*)			
β, D −10	7.059	−0.46	5.390	−1.04	9.987	+1.37	342.24	−0.05	−203.478	0.00
	(*7.092*)		(*5.447*)		(*9.852*)		(*342.40*)			
β, H −123	7.017	+0.31	5.456	−0.49	9.927	+1.60	339.79	+1.90	−203.457	0.24
	(*6.995*)		(*5.483*)		(*9.771*)		(*333.47*)			
β, D −123	7.021	+0.10	5.450	−0.42	9.927	+1.64	339.75	+1.86	−203.453	0.29
	(*7.014*)		(*5.473*)		(*9.767*)		(*333.54*)			

Notes: The corresponding measured parameters are given in parentheses. Relative errors of the calculation with respect to measurement are also listed, as well as the total DFT energy of the system (*E*) and the relative energy per stoichiometric unit (Δ*E*).

**Table 3 ijms-23-06164-t003:** The peak frequencies of the most noticeable bands in the spectra of ADAD.

Infrared Mode	Frequency [cm^−1^]	
	T = 25 °C	T = −150 °C
ν_as_OH	3508	3493
ν_s_OH	3382	3345
Toppings	3085	3081
Toppings	2971	2971
Toppings	2797	2848
Toppings	2635	2662
Toppings	2497	2516
Toppings	2288	2289
BLF	1968	1931
C=O stretch	1647	1640
	1602	1598
In-plane OH bending	1435	1439
	1271	1285
	1257	1263
Out-of-plane OH bending	1071	1107
	994	998
	758	759

**Table 4 ijms-23-06164-t004:** The peak frequencies of the most noticeable bands in the spectra of ADAD in a deuterated form.

Infrared Modes	Frequency [cm^−1^]	
	T = 25 °C	T = −150 °C
ν_as_ OD	2612	2602
ν_s_ OD	2507	2487
Toppings		2374
Toppings	2170	2179
Toppings	2047	1981
BLF	1920	1871
	1803	1776
C=O stretch	1646	1634
		1598
		1522
		1485
C-OH stretch	1352	1360
		1304
	1290	
	1185	1185
In-plane OD bending	1083	1091
	972	976
	945	951
Out-of-plane OD bending	786	811
	758	760, 750

**Table 5 ijms-23-06164-t005:** Selected harmonic modes and frequencies corresponding to vibrations of the hydrogen bond, computed for the protiated and deuterated isotopomers of the α-polymorph.

Protiated Isotopomer (H)		Deuterated Isotopomer (D)	
Frequency [cm^−1^]	Assignment	Frequency [cm^−1^]	Assignment
2260–2470	ν_OH_ (+ν_C≡C_)	1680–1810	ν_OD_
1630–1640	δ_OH_ + ν_C=O_	1530–1590	ν_OD_ + ν_C=O_
1580–1590			
1450–1470	δ_OH_ + ν_C—C_ + ν_C—O_	1070–1090	δ_OD_ + ν_C=O_
1270–1340			
1200–1210	γ_OH_	870–880	γ_OD_

**Table 6 ijms-23-06164-t006:** The assignment of the most noticeable bands in the Raman spectra of protiated ADAD.

Raman Mode	Frequency [cm^−1^]	
	T = 25 °C	T = −160 °C
ν_as_OH	3505	3489
ν_s_OH	3393	3366
νC≡C	2243	2256
Hump	1995	1995
	1803	1776
C=O coupled with water bending and in-plane OH bending	1687, 1648	1681, 1663, 1624
In-plane OH bending	1453	1448
	1328	1326
	1285	1294
		809
	791	796
	756	758
	607	610
	415	4*23*
δ C≡C-C	315	322
Phonon region		303
	286	290
	208	218
	168	171
		128
	102	97
	63	70

**Table 7 ijms-23-06164-t007:** The assignment of the most noticeable bands in the Raman spectra of deuterated ADAD.

Raman Modes	Frequency [cm^−1^]	
	T = 25 °C	T = −110 °C
ν_as_OD	2616, 2605	2602, 2592
ν_s_OD	2520	2501
νC≡C	2243	2255
C=O	1678, 1662, 1639	1674, 1639
		1589
		1516
		1489
	1359	1369
		1351
		1304
		1091
	1074	1082
		772
	753	753
	730	729
	606	606
	403	413
δ C≡C-C	312	315
Phonon region	294	296, 292
	273	271
	245	245
	198	210
		166
	98	95
	71	79
	62	68

## Data Availability

Crystallographic data are available at: www.ccdc.cam.ac.uk/conts/retrieving.html (accessed on 25 May 2022).

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
