# Peer review of "Strong Hydrogen Bonds in Acetylenedicarboxylic Acid Dihydrate"

_ijms, 2022, doi:10.3390/ijms23116164_

Round 1

Reviewer 1 Report

  1. The authors have done a great work, but it is not clear why. Why was acetylenedicarboxylic acid dihydrate chosen as the object of study? What is so remarkable about this acid? What properties? Where is it used? Why was it necessary to study its structure, solved 75 years ago, and its vibrational spectrum so carefully? Or were there purely fundamental or methodological tasks, and this material was chosen as an example? And why was it necessary to carry out quantum mechanical calculations based on accurate experimental data? It would be strange to obtain a strong discrepancy between the experimental and theoretical data under such conditions. The conclusions say nothing at all about the results of quantum mechanical calculations. What is this all for? Useful calculated parameters can be the energy of the system, the bond energy, but only if they are discussed in the text. Otherwise, such calculations are meaningless. The problem statement is not clear. It should be well described in the Introduction and the purpose of the study should be clearly stated.
  2. What is Davydov coupling? The Davydov splitting of exciton bands is known, but coupling requires an explanation.
  3. The section “Materials and methods” should be placed after the Introduction.This is more familiar to many readers.This section states that the ADAD crystals were obtained as colorless plates, and Table S4 indicates that the crystal of the β1 phase is colorless prism.What is right?The shape of the crystals of the deuterated phase should be indicated.In the same section, data on DSC should be entered.You discuss them in the text.

Do I understand correctly that the β-phase of ADAD crystals exists only at low temperatures? Or can it still be obtained at room temperature?

  1. It is not necessary to indicate in the figure captions when drawing the structure the program with which the figures were drawn. The programs are listed in the “Materials and methods” section. Ellipsoids can be drawn not only with the ORTEP program, and the structure can be drawn not only with the Mercury program. Figure captions should be simplified: "Scheme ...", "Molecular packing …", "Layer packing ...".
  2. Table captions are very cumbersome and often duplicate the information that is in the tables. The captions for Tables 1 and 2 should be shortened, limited to the first sentences. Everything else should be put in the notes to these tables. The captions for Tables 3, 4, 6 and 7 - "recorded at T=25°C and -150°C (in cm-1)" - are redundant information. The right columns in the tables should be titled “Frequency [cm–1]” instead of ADAD. In general, all columns and lines should begin with capital letters. This is generally accepted. The caption to Table 5 – “(left side) and (right side)” is also redundant information.
  3. Figure 4 shows areas C1 and C2. Explanations should be given in the figure caption. Explanations appear only towards the end of the article. It is not right.
  4. Some abbreviations need to be explained at the first mention.In general, abbreviations should be entered if they are used more than twice.

Line 45 - INS diffraction.

Line 102 - OADH.

Line 109 - QM calculations.

Line 125 - XRD.

Line 292 - ATR spectra.

Line 432 - FT Raman spectrum.

Line 637 - PED calculation.

  1. There are inaccuracies in English and slang:

Lines 13 and 14 - what are "unequal lengths of the two oxygen atoms"? Similarly

Lines 648 and 649 - "unequal lengths of both oxygens" (oxygen is always used in the singular as a chemical element).

Line 81 - What is structured OH?

Line 105 - low temperatures - this is a loose concept. Specific temperatures should be specified.

Line 121 - Specify the space group.

Lines 142 and 143 – there is an incorrect sentence: The C2-O1 bond lengths in the range of 1.286(3) to 1.299(2) A are significantly longer than the C2=O2 bonds…

Better to write: The C2-O1 bonds in the range from 1.286(3) to 1.299(2) A are significantly longer than the C2=O2 bonds…

Line 178 - What is H-bond connectivity? Maybe you meant a network of hydrogen bonds?

Line 193 - an incorrect "bond distances" expression. Either the bond lengths or the distance between the atoms.

Lines 480 and 481 – there is an incomprehensible sentence: So that C2 almost holds on to the C=O stretch.

Author Response

We would like to thank the reviewer. By applying his (her) suggestions and comments, the quality of the manuscript was greatly improved.

Answers on comments: Reviewer 1

  1. The authors have done a great work, but it is not clear why. Why was acetylenedicarboxylic acid dihydrate chosen as the object of study? What is so remarkable about this acid? What properties? Where is it used? Why was it necessary to study its structure, solved 75 years ago, and its vibrational spectrum so carefully? Or were there purely fundamental or methodological tasks, and this material was chosen as an example? And why was it necessary to carry out quantum mechanical calculations based on accurate experimental data? It would be strange to obtain a strong discrepancy between the experimental and theoretical data under such conditions. The conclusions say nothing at all about the results of quantum mechanical calculations. What is this all for? Useful calculated parameters can be the energy of the system, the bond energy, but only if they are discussed in the text. Otherwise, such calculations are meaningless. The problem statement is not clear. It should be well described in the Introduction and the purpose of the study should be clearly stated.

We have already mentioned in the introduction that strong hydrogen bonds are very interesting types of non-covalent interactions in both biology and materials science. While the dihydrate of structurally very similar oxalic acid has been extensively studied (more than 480 hits in CCDC ), amounting to dozens of scientific papers and over 480 CSD entries, this is not true for the ADAD dihydrate (only one hit in CCDC on the structure of this compound – and this is without H atoms positions). Moreover, the structural similarity on the one hand and the different O...O distance on the other gives us a rare opportunity to study the effects of hydrogen bond strength on the vibrational spectra of the two. As we wrote in the introduction, this example is almost free of side effects due to the differences in the environment.

Nevertheless the special issue is dedicated to the non-covalent interactions and among them hydrogen bond is one of the most important. This is particularly true for the class of short (strong) H-bonds for which most of the commonly observed effects are expressed in a radically different and often disproportional manner. While short H-bonds are interesting in their own right because of their intriguing features such as for example, the enormously red-shifted and broadened proton stretching bands in the infrared spectra, research on short H-bonding received strong impetus when Cleland and Kreevoy suggested that this short H-bond, now called a ‘low barrier H-bond’may play a vital role in certain enzymatic reactions (Cleland and Kreevoy, Science, 1994 264, 1887-1890). Their proposal has been widely debated and investigated both theoretically and experimentally.

The purpose of quantum calculations employed in the present study has been (i) validation of the structural model and (ii) support for assignment of infrared spectra. Regarding (i), validation is done in both ways: the (presumed) good agreement of the optimized structure with the experimental one justifies the use of the selected computational model for the subsequent evaluation of proton dynamics (i.e. interpretation of the spectra). At the same time, calculation in part confirms the validity of the diffraction-determined structure and improves the structural model, in that it provides a precise and reliable location of the H-bonded protons. This is often a challenging task for X-ray diffraction techniques, particularly in the case of short H-bonds. The choice of experimental data as starting point is evident and has been done routinely in a large number of studies. It has to be noted that, while a good match with experiment is not a rarity, is not self-evident either. Specifically, for short H-bonds, the shortness is often substantially underestimated and proper treatment of non-bonding interactions is sometimes all but trivial. Our experience with a similar system, oxalic acid dihydrate [reference 14 in the manuscript], fully supports this.

Therefore, to clarify the reviewer's comments we added into the Introduction at the beginning and at the end the following text:

…Hydrogen bonds in biological systems are crucial for structure, dynamics, and biological function. Among all types of hydrogen bonds, the short (strong) hydrogen bonds are of particular interest. The study of short hydrogen bonds was strongly motivated when Cleland and Kreevoy proposed that this type of hydrogen bonding could play a central role in certain enzymatic reactions[1]. Their proposal has been extensively studied both theoretically and experimentally. However, studies of the nature and role of strong hydrogen bonds have always been somewhat limited because of the limited number of examples available and the often difficult task of interpreting observations, such as the assignment of infrared spectral bands or precise localization of the proton in hydrogen bond. Furthermore, the exact single crystal structure of hydrogen bonded system can significantly alleviate these problems and serves as an excellent starting point for spectroscopic and theoretical studies. A representative system with short hydrogen bonds is acetylenedicarboxylic acid dihydrate (ADAD).

 .. The purpose of quantum calculations employed in the present study has been (i) validation of the structural model and (ii) support for assignment of infrared spectra. Regarding (i), validation is done in both ways: the (presumed) good agreement of the optimized structure with the experimental one justifies the use of the selected computational model for the subsequent evaluation of proton dynamics (i.e. interpretation of the spectra). At the same time, calculation in part confirms the validity of the diffraction-determined structure and improves the structural model, in that it provides a precise and reliable location of the H-bonded protons. This is often a challenging task for X-ray diffraction techniques, particularly in the case of short H-bonds. The choice of experimental data as starting point is evident and has been done routinely in a large number of studies. It has to be noted that, while a good match with experiment is not a rarity, is not self-evident either. Specifically, for short H-bonds, the shortness is often substantially underestimated and proper treatment of non-bonding interactions is sometimes all but trivial…

In the Conclusions we add:

… In the temperature range between 25 and -123 °C, there are only two polymorphic crystalline forms, α and β for both, protic and deuterated ADAD. On deuteration of ADAD only usual geometric isotope effect is observed. Deuterated α is isostructural with protic α and deuterated β with protic β structure, respectively. Neither calorimetric nor X-ray diffraction measurements support the third,  form.

… The herein employed quantum calculations are in excellent agreement with the experimental diffraction structure, thereby validating the structural model used for the evaluation of proton dynamics. Although remaining on the level of harmonic approximation, calculations have been able to enhance interpretation of the infrared spectra, fairly reproducing most of the bands related to the hydrogen bond, whereas the discrepancy between the computed and observed OH stretching band is understandable for hydrogen bond of such shortness.

  1. What is Davydov coupling? The Davydov splitting of exciton bands is known, but coupling requires an explanation.

The theory of Davydov coupling is excellently explained in the paper of Flakus and coworkers referred in journal as reference 38, of Rekik et.al. in reference 18 and at the beginning by Marechal and Witkowski in reference 23. As coupling between the low, high or excited frequency modes of hydrogen bond. To clarify the type of coupling, we added in line 75  the words

…. related to hydrogen bond …..

Moreover, the effect Davydov coupling is described and well documented with corresponding references in the Discussion , where we compare ADAD with OADH.

  1. The section “Materials and methods” should be placed after the Introduction. This is more familiar to many readers.This section states that the ADAD crystals were obtained as colorless plates, and Table S4 indicates that the crystal of the β1 phase is colorless prism.What is right?The shape of the crystals of the deuterated phase should be indicated.In the same section, data on DSC should be entered. You discuss them in the text.

We agree that it is more familiar to the reader to place the Materials and methods immediately after the Introduction. The chapter in placed after the Introduction.

Table S4 shows the shape and dimensions of the protic and deuterated crystals used for X-ray data collection. Protic crystals in solution appeared as colourless plates. Initially, we used small, tiny crystals for X-ray data collection. Later, the data collection of protic ADAD was repeated at 263 K (b1 structure) to obtain higher resolution diffraction data for better refinement of the parameters of H atoms. To obtain stronger reflections even at high diffraction angles, a larger crystal was chosen. It was obtained by cutting off part of a thick plate-shaped crystal (since the dimensions of the crystal should be smaller than 0.5 mm). In the case of deuterated ADAD, a sample considered for X-ray structure determination contained large crystals, which were also smaller in one dimension than in the other two. To obtain good diffraction data (sufficiently strong reflections), thick crystals with fairly uniform dimensions were selected, and consequently their shape was described in the table as a prism rather than a plate.
To avoid misleading conclusions, we decided to omit the estimate of the shape of a selected crystal in Table S4 and to give only the colour and dimensions of the selected crystal. In the text we have added the description of the deuterated crystals.

… in line 137 we added

Deuterated crystals were of the same colour and similar shape as protic. Large and thick crystals can grow with time in both cases.

We moved the description of the DSC experiment from Supplementary information file to the MS Material and methods paragraph as suggested by the reviewer.

3a. Do I understand correctly that the β-phase of ADAD crystals exists only at low temperatures? Or can it still be obtained at room temperature?

β-phase of ADAD exists only at low temperatures. It can't be obtained at room temperature. We made many experiments with single crystals and β-phase never appeared at room temperature.  The room temperature powder X-ray diffraction data on bulk sample confirmed that bulk consist only from a-phase. (Figure S4)

  1. It is not necessary to indicate in the figure captions when drawing the structure the program with which the figures were drawn. The programs are listed in the “Materials and methods” section. Ellipsoids can be drawn not only with the ORTEP program, and the structure can be drawn not only with the Mercury program. Figure captions should be simplified: "Scheme ...", "Molecular packing …", "Layer packing ...".

We changed the Figure captions for following figures as suggested by reviewer:

Figure 1. Atom labelling scheme of water and ADAD molecules in α (left) and β2 (right) structure. Displacement ellipsoids are at 50% probability level. (i: –x, –y, –z+2)

Figure 2: Molecular packing of α- (left) and β2- (right) structure viewed along b axis. Blue lines represent hydrogen bonds with symmetry codes *: x,y,z; i: 1–x,–1/2+y,1/2–z; ii: x,3/2–y,–1/2+z;

Figure 3. Layers which build ADAD in both polymorphic forms - α (left) and β2 (right). Above: layers viewed along b axis. Below: layers in the plane of paper. Blue lines represent hydrogen bonds. * represent H-bond connection of molecules in the same asymmetric unit, and i a symmetry relation by two fold screw axis.

  1. Table captions are very cumbersome and often duplicate the information that is in the tables. The captions for Tables 1 and 2 should be shortened, limited to the first sentences. Everything else should be put in the notes to these tables. The captions for Tables 3, 4, 6 and 7 - "recorded at T=25°C and -150°C (in cm-1)" - are redundant information. The right columns in the tables should be titled “Frequency [cm–1]” instead of ADAD. In general, all columns and lines should begin with capital letters. This is generally accepted. The caption to Table 5 – “(left side) and (right side)” is also redundant information.

As suggested we shortened the Table captions for Table 1 and Table 2 by using notes at the bottom of both tables.

The redundant information is eliminated from the Table captions (3,4,6, and 7). In these tables the right column is modified by introduction of the …Frequency in cm-1… as suggested.

Expressions Left side and Right side are eliminated from the Table 5 caption.

All columns and lines now begin with capital letters.

  1. Figure 4 shows areas C1 and C2. Explanations should be given in the figure caption. Explanations appear only towards the end of the article. It is not right.

We added the short explanation in the Figure caption

…C1 and C2 represent high and low frequency regions of broad OH stretch…

  1. Some abbreviations need to be explained at the first mention. In general, abbreviations should be entered if they are used more than twice.

Line 45 - INS diffraction.

Line 102 - OADH.

Line 109 - QM calculations.

Line 125 - XRD.

Line 292 - ATR spectra.

Line 432 - FT Raman spectrum.

Line 637 - PED calculation.

All abbreviations were corrected and explained at first appearance.

  1. There are inaccuracies in English and slang:

Lines 13 and 14 - what are "unequal lengths of the two oxygen atoms"? Similarly

The sentence is changed with:

… clearly shows different bond lengths of the two oxygen atoms with respect to the carbon atom in the carboxyl group, ….

Lines 648 and 649 - "unequal lengths of both oxygens" (oxygen is always used in the singular as a chemical element).

The sentence is changed with:

… demonstrates differences in bond length of both oxygen atoms with…

Line 81 - What is structured OH?

The sentence is changed with

…. structured OH stretching band….

It relates to the complex structure of OH stretching band

Line 105 - low temperatures - this is a loose concept. Specific temperatures should be specified.

We agree, the new sentence is:

… and the crystal structures of the polymorphic modification at 4.9°C and near -40°C were determined by X-ray single crystal structure analysis

Line 121 - Specify the space group.

A sentence: Both polymorphs have a monoclinic unit cell and the same space group (14).

..is replaced with: Both polymorphs have a monoclinic unit cell and the same space group P21/c no. 14.

Lines 142 and 143 – there is an incorrect sentence: The C2-O1 bond lengths in the range of 1.286(3) to 1.299(2) A are significantly longer than the C2=O2 bonds…

Better to write: The C2-O1 bonds in the range from 1.286(3) to 1.299(2) A are significantly longer than the C2=O2 bonds…

We changed the old sentence with the proposed one.

Line 178 - What is H-bond connectivity? Maybe you meant a network of hydrogen bonds?

The sentence is changed to:

… while the hydrogen bonds are not broken. The connectivity of the hydrogen bonds within the network remains the same.

Line 193 - an incorrect "bond distances" expression. Either the bond lengths or the distance between the atoms.

Bond lengths is now used.

Lines 480 and 481 – there is an incomprehensible sentence: So that C2 almost holds on to the C=O stretch.

This sentence is removed.

Reviewer 2 Report

The authors carefully characterized and reexamined the crystal structure and vibrational structure of acetylenedicarboxylic acid dihydrate by providing fine details, and used QM calculations to further support the experimental results. They compared the structure and vibrational spectra between ADAD and oxalic acid dihydrate, and found some interesting discoveries. Protonated and deuterated ADAD were also compared in terms of Xray structure and vibrational spectra measurement at different temperatures. The writing is scientifically sound and clear. Therefore I recommend it to be published after minor revisions:

  1. The authors did a temperature dependent studies on ADAD and deuterated ADAD in terms of IR and Raman. However, in different measurements, the temperatures used is not consistent. In figure 4, it's 25 C and -150 C. While in figure 6 and 7. The temperatures are different. Can the authors explain why?
  2. The appearance of "materials and methods" is not reasonable. The authors may want to rearrange it and put it either after the intro part or in the end.

Author Response

We would like to thank the reviewer. By applying his (her) suggestions and comments, the quality of the manuscript was greatly improved.

  1. The authors did a temperature dependent studies on ADAD and deuterated ADAD in terms of IR and Raman. However, in different measurements, the temperatures used is not consistent. In figure 4, it's 25 C and -150 C. While in figure 6 and 7. The temperatures are different. Can the authors explain why?

The explanation lies in the fact that we are forced to use three different temperature cells with different final temperatures. For all infrared measurements, we used a low-temperature cell that can only reach -110 °C, while Raman spectra were recorded with two different low-temperature cells that cannot be removed from the spectrometers. Therefore, we used the lowest temperature reached by three different low-temperature cells.

  1. The appearance of "materials and methods" is not reasonable. The authors may want to rearrange it and put it either after the intro part or in the end.

We moved this section after Introduction.

Reviewer 3 Report

The manuscript “Strong hydrogen bonds in acetylene dicarboxylic acid dihydrate” by Grdadolnik et al. deals with very fine details of the vibrational dynamics of acetylene dicarboxylic acid dihydrate  (ADAD) on the base of several crystal structures, including deuterated molecules. The theoretical studies are thorough, although in my opinion the text is too long and some tedious. I think this is probably the result of an unclear objective beyond a purely theoretical interest of the authors, as I cannot see any practical objective, or even interest in such molecule. So I think that this is a serious inconvenience to arise the interest of the readers of IJMS. In fact, in my humble opinion, the article is not very appropriate for this journal, and probably more adequate for journal devoted to spectral and/or theoretical studies

According to authors, some of the interest of this study appears to be the comparison of the effects of deuteration of ADAD, to be compared with that of oxalic acid, by the way, a similar molecule whose vibrational spectra have been studied by some of the authors (ref. 6,7,13,17). but not why this molecule deserves such detailed study, or some other potential interest of this particular molecule.

Most of these studies are based on six crystal structures of ADAD (CCDC 2160304-2160309), studied by single crystal X-ray diffraction and very recently deposited. This useful technique, is however, not the most appropriate for studying such subtle details of the H-bonding, especially if data of quality is not excellent. As crystallographer, I regret to say that the data presented are not nowadays acceptable, as these cifs and the programs used for such studies (SIR 97 and especially SHEXL 97) are absolutely old-fashioned. I have to indicate that much modern versions (SHELXL Version 2018/3) of these programs are absolutely free, so that this aspect is unacceptable. Furthermore,  the cifs so generated are consequently truncated, and its quality cannot be deeply checked as recommended nowadays.

I also miss some comparison with other crystal structures of the acetylene dicarboxylic acid or containing it, as for instance, a significant article for this manuscript as https://doi.org/10.1021/acs.cgd.7b01338.

Author Response

We would like to thank the reviewer. By applying his (her) suggestions and comments, the quality of the manuscript was greatly improved.

The manuscript “Strong hydrogen bonds in acetylene dicarboxylic acid dihydrate” by Grdadolnik et al. deals with very fine details of the vibrational dynamics of acetylene dicarboxylic acid dihydrate (ADAD) on the base of several crystal structures, including deuterated molecules. The theoretical studies are thorough, although in my opinion the text is too long and some tedious. I think this is probably the result of an unclear objective beyond a purely theoretical interest of the authors, as I cannot see any practical objective, or even interest in such molecule. So I think that this is a serious inconvenience to arise the interest of the readers of IJMS. In fact, in my humble opinion, the article is not very appropriate for this journal, and probably more adequate for journal devoted to spectral and/or theoretical studies

According to authors, some of the interest of this study appears to be the comparison of the effects of deuteration of ADAD, to be compared with that of oxalic acid, by the way, a similar molecule whose vibrational spectra have been studied by some of the authors (ref. 6,7,13,17). but not why this molecule deserves such detailed study, or some other potential interest of this particular molecule.

The unique and intriguing properties of H-bonding have been long subject of profound investigations using a wide array of experimental and computational research techniques. Despite the impressive amount of the information that has been collected, many characteristics of the H-bond remain poorly understood. This is particularly true for the class of short (strong) H-bonds for which most of the commonly observed effects are expressed in a radically different and often disproportional manner. While short H-bonds are interesting in their own right because of their intriguing features such as for example, the enormously red-shifted and broadened proton stretching bands in the infrared spectra, research on short H-bonding received strong impetus when Cleland and Kreevoy suggested that this short H-bond, now called a ‘low barrier H-bond’may play a vital role in certain enzymatic reactions (Cleland and Kreevoy, Science, 1994 264, 1887-1890). Their proposal has been widely debated and investigated both theoretically and experimentally. Nonetheless studies of the nature and role of short H-bonding have always been somewhat limited on account of a shortage of examples and also because of difficulties in the interpretation of observables, which can often be a challenging task, such as finding the precise location of the proton, or assignment of infrared spectral bands.

We strongly believe that our comprehensive study of ADAD, which may represent the model system for short hydrogen bonds, will be of interest not only to researchers in the field of biological research but also to a broader community, including researchers in the field of material science.

To emphasise the importance of these type or research we change the Introduction and Conclusions.

…added paragraphs to  the Introduction and Conclusions ::

..begining of the Introduction

…Hydrogen bonds in biological systems are crucial for structure, dynamics, and biological function. Among all types of hydrogen bonds, the short (strong) hydrogen bonds are of particular interest. The study of short hydrogen bonds was strongly motivated when Cleland and Kreevoy proposed that this type of hydrogen bonding could play a central role in certain enzymatic reactions[1]. Their proposal has been extensively studied both theoretically and experimentally. However, studies of the nature and role of strong hydrogen bonds have always been somewhat limited because of the limited number of examples available and the often difficult task of interpreting observations, such as the assignment of infrared spectral bands or precise localization of the proton in hydrogen bond. Furthermore, the exact single crystal structure of hydrogen bonded system can significantly alleviate these problems and serves as an excellent starting point for spectroscopic and theoretical studies. A representative system with short hydrogen bonds is acetylenedicarboxylic acid dihydrate (ADAD).

…end of Introduction

 .. The purpose of quantum calculations employed in the present study has been (i) validation of the structural model and (ii) support for assignment of infrared spectra. Regarding (i), validation is done in both ways: the (presumed) good agreement of the optimized structure with the experimental one justifies the use of the selected computational model for the subsequent evaluation of proton dynamics (i.e. interpretation of the spectra). At the same time, calculation in part confirms the validity of the diffraction-determined structure and improves the structural model, in that it provides a precise and reliable location of the H-bonded protons. This is often a challenging task for X-ray diffraction techniques, particularly in the case of short H-bonds. The choice of experimental data as starting point is evident and has been done routinely in a large number of studies. It has to be noted that, while a good match with experiment is not a rarity, is not self-evident either. Specifically, for short H-bonds, the shortness is often substantially underestimated and proper treatment of non-bonding interactions is sometimes all but trivial…

Add to the Conclusion:

… In the temperature range between 25 and -123 °C, there are only two polymorphic crystalline forms, α and β for both, protic and deuterated ADAD. On deuteration of ADAD only usual geometric isotope effect is observed. Deuterated α is isostructural with protic α and deuterated β with protic β structure, respectively. Neither calorimetric nor X-ray diffraction measurements support the third,  form.

… The herein employed quantum calculations are in excellent agreement with the experimental diffraction structure, thereby validating the structural model used for the evaluation of proton dynamics. Although remaining on the level of harmonic approximation, calculations have been able to enhance interpretation of the infrared spectra, fairly reproducing most of the bands related to the hydrogen bond, whereas the discrepancy between the computed and observed OH stretching band is understandable for hydrogen bond of such shortness.

 Most of these studies are based on six crystal structures of ADAD (CCDC 2160304-2160309), studied by single crystal X-ray diffraction and very recently deposited. This useful technique, is however, not the most appropriate for studying such subtle details of the H-bonding, especially if data of quality is not excellent. As crystallographer, I regret to say that the data presented are not nowadays acceptable, as these cifs and the programs used for such studies (SIR 97 and especially SHEXL 97) are absolutely old-fashioned. I have to indicate that much modern versions (SHELXL Version 2018/3) of these programs are absolutely free, so that this aspect is unacceptable. Furthermore, the cifs so generated are consequently truncated, and its quality cannot be deeply checked as recommended nowadays.

Crystallographic data are of a very good quality according to all crystallographic standards for X-ray structure determination: there is a good agreement between calculated and measured magnitude of structure factors (intensities) and between symmetry equivalent reflections (low R and Rint factors). Final maximal and minimal (absolute) values in residual electron density are small, standard uncertainties (ESD) are small enough, and shifts/ESD are small. All these values are given in Table S4 and in the deposited CIF files. Deposited were both, structural .CIF files and also reflection data (.FCF files). Crystallographic data were successfully checked prior deposition via standard checkcif procedure (https://checkcif.iucr.org).

Since the reviewer 3 wanted to check structure determination using CIF files in format generated by SHELXL Version 2018/3, we generated new version of CIF files (for all six structures) by the refinement of structures with SHELXL Version 2018/3. As expected, structures practically did not change after their refinement with SHELXL 2018/3, since versions of SHELXL97 and SHELXL2018/3 use the same scattering factors for C, H and O atoms and the same mathematical procedure of refinement. There were only few small changes in the numbers ​​in the last place of accuracy as a result of rounding of computer, as the numbers in the computer are written to a limited number of places. These changes are inserted into the tables (S4 and S5) in the revised version of the manuscript and don’t affect the discussion and conclusions in any way. We also updated deposited structural data with SHELXL Version 2018/3 CIF files.

 I also miss some comparison with other crystal structures of the acetylene dicarboxylic acid or containing it, as for instance, a significant article for this manuscript as https://doi.org/10.1021/acs.cgd.7b01338.

We agree that acetylenedicarboxylic acid is an extremely interesting molecule with some unique properties. However, in this study we have focused on the occurrence of the strong (short) hydrogen bond between the acetylenedicarboxylic acid and the hydrating water molecule, which is similar to that found in oxalic acid dihydrate. We have taken this rare opportunity to study two structurally very similar systems with short hydrogen bonds of different strengths. And that's why we need to do a detailed study at ADAD. Since the manuscript is already very long, an additional comparison with other systems containing the acetylenedicarboxylic acid was not mentioned.

Round 2

Reviewer 1 Report

The authors have significantly improved the manuscript. However, some comments remained, some of which, unfortunately, were missed in the first round.

  1. In the Materials and Methods section, the authors refer to Table 1, which gives Details on crystal data, data collection and structure refinement.

Apparently, the authors meant Table S4, because Table 1 presents the results of DFT calculations in comparison with experimental data. I recommend moving Table S4 into the text body, because it is important, especially since you are talking about an accurate experiment.

  1. In crystallography, there is a term – atomic displacement parameters (ADP), so in line 167 it is better to write: with anisotropic displacement parameters for…
  2. Lines 170 and 171: Drawings of the structures were produced using ORTEPIII [36] and Mercury [37].It's easier to write: The structures were drawn using ORTEPIII [36] and Mercury [37].
  3. Line 193: Bad statement "Each of the X-ray recorded structure".

Better to write: Each of the structure studied by X-ray diffraction analysis

  1. Line 108: low temperatures have not been corrected.
  2. Lines 291 and 292: The connectivity of the hydrogen bonds is a tautology. I suggest two options to choose from.

The connectivity of hydrogen atoms that form the network remains the same.

OR

The network of hydrogen bonds remains the same.

  1. Designations must be uniform throughout the article. If you enclose units of measurement in square brackets, then this must be done everywhere. The right columns in Tables 3, 4, 6 and 7 should be titled “Frequency [cm–1]” (rather than Frequency in cm–1).

Author Response

Referee 1

Once again, we would like to thank you for very constructive comments, all of which were accepted as follows:

The authors have significantly improved the manuscript. However, some comments remained, some of which, unfortunately, were missed in the first round.

We are very pleased to have succeeded in improving the quality of our manuscript.

  1. In the Materials and Methods section, the authors refer to Table 1, which gives Details on crystal data, data collection and structure refinement.

Apparently, the authors meant Table S4, because Table 1 presents the results of DFT calculations in comparison with experimental data. I recommend moving Table S4 into the text body, because it is important, especially since you are talking about an accurate experiment.

The Table 1 is really not the right one. The reader is directed to SI and Table S4

  1. In crystallography, there is a term – atomic displacement parameters (ADP), so in line 167 it is better to write: with anisotropic displacement parameters for…

»temperature« was deleted

  1. Lines 170 and 171: Drawings of the structures were produced using ORTEPIII [36] and Mercury [37].It's easier to write: The structures were drawn using ORTEPIII [36] and Mercury [37].

The statement is changed by following the reviewer's suggestion

  1. Line 193: Bad statement "Each of the X-ray recorded structure".

Better to write: Each of the structure studied by X-ray diffraction analysis

The statement is changed by following the reviewer's suggestion.

  1. Line 108: low temperatures have not been corrected.

… low temperature…  is substituted with … 20°C, -10°C and -123°C….

  1. Lines 291 and 292: The connectivity of the hydrogen bonds is a tautology. I suggest two options to choose from.

The connectivity of hydrogen atoms that form the network remains the same.

OR

The network of hydrogen bonds remains the same.

The statement is changed by following the reviewer's suggestion

The network of hydrogen bonds remains the same.

  1. Designations must be uniform throughout the article. If you enclose units of measurement in square brackets, then this must be done everywhere. The right columns in Tables 3, 4, 6 and 7 should be titled “Frequency [cm–1]” (rather than Frequency in cm–1).

All »Frequency in cm–1« is substituted with  “Frequency [cm–1]”

Last but not least, we would like to reiterate the aim of the present study.
Briefly, our motive for studying ADAD can also be summarized as follows. OADH and ADAD are of interest to spectroscopists because of their strong H-bonds. Much has been written about OADH, but little about ADAD. An article from 70 years ago reported that two phase transitions occur when ADAD is cooled. However, the spectra of IR showed no significant changes upon cooling. The only structure determination of ADAD was made 75 years ago using film measurements without determining the position of H atoms and large R factors, so it was reasonable to make a redetermination of the structure at room temperature. At the same time, we wanted to check whether the X-ray structural analysis confirmed the phase transitions and what differences in the structure occurred. Using the parameters of the unit cell, we were able to determine very quickly that only one and not two phase transitions occur when cooling from 20 degrees to -123 °C. At the same time, it was not immediately obvious from the determination itself what happens to the structure during the phase transition. An automatic program for determining the unit cell on the diffractometer selects a base cell that is closer to 90 ° (a=6.995, b=5.483, c=9.057Å, β=106.3) and the asymmetric unit was also chosen differently, so the α- and β-structures did not look similar at all at first. It was necessary to find a suitable transformation of the coordinate system and the origin so that we could compare the structures and find out what happens during the phase transition. Only then did we see that only the mutual orientation of the molecules within the layers changes during the phase transition, while the hydrogen bonds are not broken. All the findings were supported by theoretical quantum chemical calculations. It is gratifying that their results agree with experimental (spectroscopic, crystallographic, and DSC) observations.
In addition, we wanted to test whether isotopic polymorphism occurs after deuteration of ADAD.

We hope that we have significantly improved our manuscript with strict attention to your valuable advice, comments and suggestions.

Reviewer 3 Report

I satisfied with many of the changes made, especially with the use of modern programs to solve the crystal structures. However I am sorry to say that I still keep most of my opinions about the manuscript, and despite my sincere interest in H-bonding, I am sorry to say that the manuscript does not still catch my interest. I heve read the covering letter and the introduction, but the reasons to choose this particular and predictable molecule are not clear to me.

The main reason for my not satisfactory impression may  probably be related to the disapointing study on the crystal structures, firstly with old programs, and now I have checked that it does not appear thorough enough to me. Among other reasons, because some of these structures, when re-refined, show the possibility of bifurcated H bonds, something not considered by authors. Just only one example. For instance, for 6, I have found that after re-refining the H bonding scheme corresponds to:

 D-H                    d(D-H)   d(H..A)   <DHA    d(D..A)    A
 O1-D1                   0.909    1.648   176.71    2.555    O3 
 O3-D2                   0.884    2.166   141.43    2.910    O2 [ -x+1, y+1/2, -z+3/2 ]
 O3-D2                   0.884    2.432   130.25    3.078    O3 [ -x+1, y+1/2, -z+3/2 ]
 O3-D3                   0.955    1.832   173.30    2.783    O2 [ x, -y+3/2, z-1/2 ]

This flaw is key for me, and leads me to reject the paper, as I realise that it would be to rethink it almost completely.

In any case, the use of crystal nomenclature without italics is not careful and  should be revised.

Author Response

Dear reviewer,

We would like to thank you for very constructive comments, although there are still some mutual disagreements. And we would like to sincerely eliminate them.

 I satisfied with many of the changes made, especially with the use of modern programs to solve the crystal structures. However, I am sorry to say that I still keep most of my opinions about the manuscript, and despite my sincere interest in H-bonding, I am sorry to say that the manuscript does not still catch my interest. I have read the covering letter and the introduction, but the reasons to choose this particular and predictable molecule are not clear to me.

The study of short hydrogen bond presented in the manuscript was originally based on the infrared study of systems with short hydrogen bonds. ADAD is a nearly ideal crystal in which this short hydrogen bond occurs and, most importantly, ADAD has a very limited number of internal vibrations that can (and generally does) shield the broad absorption of the OH stretching vibration from the short hydrogen bond. When we found that there was only one poor X-ray structure, we decided to study it again in detail. We must emphasise that the X-ray structure is a prerequisite for accurate vibrational analysis. In addition, our primary goal was to compare the structurally similar oxalic acid dihydrates and ADAD, observing in the former an ionic defect leading to polaron formation, the need for a structural rearrangement around a short hydrogen bond between the water molecule and the carboxyl group. When we found that the original X-ray structure was different from ours, we decided to publish it in the context presented in our manuscript.

A few years ago we found out some patterns in the vibrational spectra of oxalic acid dihydrate that are somehow different than expected for crystals with hydrogen bonds with typical O...O distances between 2.5-2.6Å [1-6]. And in ADAD we found a crystal with similar short hydrogen bonds. It is interesting to note that both oxalic acid dihydrate and perfluorated fatty acid monohydrates generally have very similar (but not identical) vibrational spectra characterising the strong hydrogen bonding, but the reason(s) for their unique shape is (are) not yet clear.
In all these crystals we also found the OH stretchings of the weak hydrogen bonds, which are also bifurcated. But the vibrational behaviour of the hydrogen bonds is completely predictable and does not show any unusual patterns. Moreover, the frequency and bandshape can be well reproduced by calculations, implying the relative “simplicity” of the hydrogen bond potential. This is of course not true for the strong hydrogen bonds in the same systems.  This is the only reason why we have not been interested in the weak hydrogen bonds in this manuscript, as well as in others dealing with oxalic acid dihydrate and perfluorated fatty acid monohydrates.

The main reason for my not satisfactory impression may  probably be related to the disapointing study on the crystal structures, firstly with old programs, and now I have checked that it does not appear thorough enough to me. Among other reasons, because some of these structures, when re-refined, show the possibility of bifurcated H bonds, something not considered by authors. Just only one example. For instance, for 6, I have found that after re-refining the H bonding scheme corresponds to:

 D-H                    d(D-H)   d(H..A)   <DHA    d(D..A)    A
 O1-D1                   0.909    1.648   176.71    2.555    O3 
 O3-D2                   0.884    2.166   141.43    2.910    O2 [ -x+1, y+1/2, -z+3/2 ]
 O3-D2                   0.884    2.432   130.25    3.078    O3 [ -x+1, y+1/2, -z+3/2 ]
 O3-D3                   0.955    1.832   173.30    2.783    O2 [ x, -y+3/2, z-1/2 ]

This flaw is key for me, and leads me to reject the paper, as I realise that it would be to rethink it almost completely.

In any case, the use of crystal nomenclature without italics is not careful and  should be revised.

It is true that in all six crystal structures there is within spiral chains an additional stabilizing O3… O3i contact in the range of 3.088(3)- 3.137(2) Å with O3-H2…O3i angle of ~ 125°  (in range of 120-126°), due to very weak hydrogen bond between neighbouring water molecules. Since O3 of water molecule is through H2 also a donor of a hydrogen bond also to O2i from carboxyl group (stabilizing O2… O3i contact in the range of 2.8513(18)-2.909(2) Å with O3-H2…O3i angle of ~ 146° (in range of 141-154°)), such three-centre hydrogen bond can be described as bifurcated. These hydrogen bonds we had observed before submitting manuscript to the publication. (These contacts and bifurcated bonds did not show after re-refinemen.  Re-refinement did not change structures). We decided not to mention these contacts for two reasons: These additional H-bonds are very weak and we discuss in our paper strong hydrogen bonding. Such weak bifurcated Hydrogen bonds are present also in both polymorph modification of oxalic acid dihydrate and not mentioned in (numerous) papers which describe their structures and hydrogen bonds in particular the strong ones.

However, we agree with the reviewer that such geometry of three-centre hydrogen bond can be described as bifurcated hydrogen bond, so we added a text on this observation in manuscript. We added also geometrical parameters of these weak hydrogen bonds in the table S6, presented in the Supplementary information. We did not make new Figures which would include dotted lines for these weak contacts because then the more important, stronger hydrogen bonds (which only affect IR spectra) would be less visible. But if the reviewer or editor deems it necessary to draw them also, we will prepare new figures.

We also revised crystallographic nomenclature which refers to italics.

Last but not least, we would like to reiterate the aim of the present study.
Briefly, our motive for studying ADAD can also be summarized as follows. OADH and ADAD are of interest to spectroscopists because of their strong H-bonds. Much has been written about OADH, but little about ADAD. An article from 70 years ago reported that two phase transitions occur when ADAD is cooled. However, the spectra of IR showed no significant changes upon cooling. The only structure determination of ADAD was made 75 years ago using film measurements without determining the position of H atoms and large R factors, so it was reasonable to make a redetermination of the structure at room temperature. At the same time, we wanted to check whether the X-ray structural analysis confirmed the phase transitions and what differences in the structure occurred. Using the parameters of the unit cell, we were able to determine very quickly that only one and not two phase transitions occur when cooling from 20 degrees to -123 °C. At the same time, it was not immediately obvious from the determination itself what happens to the structure during the phase transition. An automatic program for determining the unit cell on the diffractometer selects a base cell that is closer to 90 ° (a=6.995, b=5.483, c=9.057Å, β=106.3) and the asymmetric unit was also chosen differently, so the α- and β-structures did not look similar at all at first. It was necessary to find a suitable transformation of the coordinate system and the origin so that we could compare the structures and find out what happens during the phase transition. Only then did we see that only the mutual orientation of the molecules within the layers changes during the phase transition, while the hydrogen bonds are not broken. All the findings were supported by theoretical quantum chemical calculations. It is gratifying that their results agree with experimental (spectroscopic, crystallographic, and DSC) observations.
In addition, we wanted to test whether isotopic polymorphism occurs after deuteration of ADAD.

We hope that we have significantly improved our manuscript with strict attention to your valuable advice, comments and suggestions.

References:

  1. Novak, U.; Grdadolnik, J. Infrared spectra of hydrogen bond network in lamellar perfluorocarboxylic acid monohydrates. Spectrochimica Acta Part A: Molecular and Biomolecular Spectroscopy 2021, 253, 119551, doi:https://doi.org/10.1016/j.saa.2021.119551.
  2. Ivanovski, V.; Mayerhöfer, T.G.; Stare, J.; Gunde, M.K.; Grdadolnik, J. Analysis of the polarized IR reflectance spectra of the monoclinic α-oxalic acid dihydrate. Spectrochimica Acta Part A: Molecular and Biomolecular Spectroscopy 2019, 218, 1-8, doi:https://doi.org/10.1016/j.saa.2019.03.094.
  3. Mohaček-Grošev, V.; Grdadolnik, J.; Hadži, D. Evidence of Polaron Excitations in Low Temperature Raman Spectra of Oxalic Acid Dihydrate. The Journal of Physical Chemistry A 2016, 120, 2789-2796.
  4. Mohaček Grošev, M.; Grdadolnik, J.; Stare, J.; Hadži, D. Identification of Hydrogen Bond Modes in Polarized Raman Spectra of Single crystala of a-Oxalic Acis Dihydrate. J. Raman Spectroscopy 2009, 40, 1605-1614.
  5. Stare, J.; Hadži, D. Cooperativity Assisted Shortening of Hydrogen Bonds in Crystalline Oxalic Acid Dihydrate : DFT and NBO Model Studies,. Journal of chemical theory and computation, 2014, 10, 1817-1823.
  6. Stare, J.; Meden, A.; Hadži, D. Structure Determination by Joint Effort of X-ray Powder Diffraction and Quantum Calculations: Crystal Structure and Short Hydrogen Bonding in Pentadecafluorooctanoic Acid Hydrate. Croat. Chem. Acta 2018, 91, 209-220.

Round 3

Reviewer 3 Report

I can see that the authors demonstrate an extraordinary interest only in the strong H bonds of this particular dihydrate compound, more than in the strong H bond that the much more simple anhydrous acid also present. By the way, a compound  with up to thirteen crystal structures recently solved (in different conditions) available in the bibliography since 2018, as I have indicated in my first revision (https://doi.org/10.1021/acs.cgd.7b01338).

The response was that ; "However, in this study we have focused on the occurrence of the strong (short) hydrogen bond between the acetylenedicarboxylic acid and the hydrating water molecule, which is similar to that found in oxalic acid dihydrate. We have taken this rare opportunity to study two structurally very similar systems with short hydrogen bonds of different strengths. And that's why we need to do a detailed study at ADAD. Since the manuscript is already very long, an additional comparison with other systems containing the acetylenedicarboxylic acid was not mentioned.

I agree that, as I have already commented, the article is already very long (but not finished according to the authors), so rather than a repetitive work comparing with such a similar system, in my point of view, comparing it with the simple anhydrous system would increase the interest of this work. Likiwise, the length of the paper could be reduced by presenting part of the studies as supplementary material.

Author Response

We have followed the reviewer's recommendation and add the description of the hydrogen bonding structure of an anhydrous acetylenedicarboxylic acid. The description is announced in the manuscript with the appropriate reference to the original paper and described in detail in the SI.
The description of the distances in anhydrous acetylenedicarboxylic acid shows the differences in hydrogen bond distances between anhydrous and hydrated acetylenedicarboxylic acid.

The interatomic distances between oxygen atoms indicate that in this system we have hydrogen bonds of moderate strength and with a completely different and predictable frequency and shape of the OH stretching characteristic of acid dimers (see, e.g., Y. Marechal, The hydrogen bond and the water molecule, Elsevier, 2007).

In the MS we added:

The comparison of these geometrical parameters with those in the structure of anhydrous acetylenedicarboxylic acid [53]  is given in SI.

In the SI we added the figure and description of hydrogen bonds found between the anhydrous acetylenedicarboxylic acid molecules.

Followed by the figure and text added in the SI:

Figure S10: Presentation of molecules of acetylenedicarboxylic acid in anhydrous structure (ADCA) [Delori et.al.]. The lengths of bonds between carbon atoms in ADCA [(C2ºC3 1.188, C1-C2 1.454 and C3-C4 1.455 Å) are similar to that of ADAD. On the other hand, the bond lengths C1»O2 1.262, C1»O3 1.245, C4»O1 1.259 and C4»O4 1.246 Å and H atoms positions with 50% occupancy reflect that carboxyl groups in ADCA are disordered which is not the case in ADAD. The conformation of acid molecules in anhydrous structure also differs from that of hydrated. In ADAD molecules are centrosymmetric and nearly planar while in ADCA molecules are asymmetric and twisted with torsion angle O2-C1-C4-O4 of » 60°. Hydrogen bonds between acid and water molecules in ADAD are significantly stronger then hydrogen bonds between acid molecules in ADCA (O3…Oi 2.656 and O2…O4i 2.694 Å, i: -1/2+x,1.5-y,1/2+z). The structure of ADCA was determined also at high pressures from 0.18 up to 5.2 GPa) [Delori et.al.]. At high pressure the structure of ADCA remains to be disordered, with twisted molecules and hydrogen bonds of moderate strength.

Delori et. al. Reaction of Acetylenedicarboxylic Acid Made Easy: High-Pressur Route for Polymerization Cryst. Growth Des. 2018, 18, 1425−1431
